# Conserved subcortical processing in visuo-vestibular gaze control

Tobias Wibble [1,2], Tony Pansell [2], Sten Grillner [1] & Juan Pérez-Fernández [1,3] ✉

Gaze stabilization compensates for movements of the head or external environment to minimize image blurring. Multisensory information stabilizes the scene on the retina via the vestibulo-ocular (VOR) and optokinetic (OKR) reflexes. While the organization of neuronal circuits underlying VOR is well-described across vertebrates, less is known about the contribution and evolution of the OKR and the basic structures allowing visuo-vestibular integration. To analyze these neuronal pathways underlying visuo-vestibular integration, we developed a setup using a lamprey eye-brain-labyrinth preparation, which allowed coordinating electrophysiological recordings, vestibular stimulation with a moving platform, and visual stimulation via screens. Lampreys exhibit robust visuo-vestibular integration, with optokinetic information processed in the pretectum that can be downregulated from tectum. Visual and vestibular inputs are integrated at several subcortical levels. Additionally, saccades are present in the form of nystagmus. Thus, all basic components of the visuo-vestibular control of gaze were present already at the dawn of vertebrate evolution.

The appearance of image-forming eyes enabled animals to use vision for advanced behavioral repertoires that required mechanisms to stabilize the world in the retina to prevent image degradation[1]. In vertebrates, visual stabilization is achieved via two reflexes thought to have evolved in parallel[1,2]. Head movements evoke compensatory shifts of the eyes mediated by vestibular signals through the vestibulo-ocular reflex (VOR). In the optokinetic reflex (OKR), retinal optic flow is fed back to the eye muscles, generating compensatory movements to stabilize the image. Complementary somatosensory inputs also contribute to image stabilization[3,4]. In addition corollary discharges allow the brain to distinguish passive from actively generated head movements[5]. The VOR and OKR act together to compensate for the disturbance generated by head movements, and contribute to robust eye movement control. As the velocity of the head movement increases the vestibular contribution becomes dominating[6,7].

Despite the impact that visual motion has on gaze stabilization, the contribution of this system has been the least studied as compared to the vestibular system. The neuronal substrate of OKR lies in the pretectal area/accessory optic system in the vertebrates studied[8,9], but the neuronal mechanisms by which large-field visual motion generates the OKR and its evolutionary origin are poorly understood. The VOR, on the other hand, appeared very early during vertebrate evolution and, although different species-specific configurations may exist, the disynaptic circuit that converts vestibular information in the appropriate compensatory eye movement is present in all vertebrates[10].

Gaze-stabilization represents a primordial motor command, few studies have however addressed the contribution resulting from VOR and OKR interactions. Visuo-vestibular integration has been analyzed mostly at the level of cortex and cerebellum[11], rather than focusing on the subcortical processing fundamental to the VOR/OKR. We here analyze stabilizing eye movements in the lamprey, the oldest extant vertebrate, and the neuronal pathways responsible for multisensory integration. Although with small differences in retinal organization when compared to gnathostomes[12], lampreys have well-developed

[1]The Department of Neuroscience, Karolinska Institutet, Stockholm, Sweden. [2]The Department of Clinical Neuroscience, Marianne Bernadotte Centrum, St: Erik's Eye Hospital, Karolinska Institutet, Stockholm, Sweden. [3]CINBIO, Universidade de Vigo, Neurocircuits Group, Campus universitario Lagoas, Marcosende 36310 Vigo, Spain. ✉e-mail: jperezf@uvigo.es

image-forming camera eyes, and the organization of eye muscles and motor nuclei is remarkably similar to other vertebrates[1,13–15]. They exhibit VOR[16] and a well-developed vestibular system, featuring a labyrinth with two semicircular canals (anterior and posterior) considered homologous to their gnathostome counterparts, while lacking a lateral horizontal canal[17–19]. However, lampreys have a horizontal duct system that provides vestibular information also in the yaw plane, which seems to have evolved in parallel to the gnathostome horizontal canal[17–19]. The lamprey labyrinth is therefore capable of generating compensatory eye and body movements in the yaw plane using the same brain circuit architecture responsible for roll and pitch rotations[20]. Whether lampreys have optokinetic eye movements is still unclear. The fact that they have retinotopic representation in both tectum and visual cortex, and a precise control of eye movements from tectum suggests that vision may contribute to gaze stabilization[21–26].

In addition to the compensatory slow eye movement, VOR/OKR responses feature nystagmus beats, quick resetting of eye position that takes place when the eyes reach the limit of their range within the orbit. These fast reset movements are considered to be the origin of saccades and constitute, together with VOR and OKR, the blueprint from which all types of eye movements derive[1,27]. Although eye movements can be evoked by electrically stimulating the optic tectum or the motor area in pallium, as well as by presenting visual stimuli[21,24,28], it is yet not known whether saccades are present or not.

In this study, we show that the OKR is present in the lamprey, and that its primary visual processing takes place in the pretectum, which forwards information to the oculomotor and vestibular nuclei. Moreover, the robust additive effect between the OKR and VOR reported in mammals was present already in the lamprey, and visual and vestibular signals are integrated at several subcortical levels, indicating that visuo-vestibular gaze-stabilization does not require cortical or cerebellar processing, although can be down-regulated from tectum. The lamprey also exhibits clear nystagmus beats, showing that all stabilizing types of eye movements were present already when the lamprey diverged from the evolutionary line leading to mammals.

## Results

### A visuo-vestibular platform to analyze the interaction between VOR and OKR

To confirm that lampreys exhibit VOR[16,29], we performed behavioral experiments, video tracking eye movements in response to vestibular stimulation. Intact animals were placed in a transparent tube filled with cold water (Fig. 1a). Eye movements were tracked using the DeepLabCut software[30] (Fig. 1b). Vestibular stimulation gave rise to prominent compensatory eye movements in the three planes (roll, pitch and yaw; $N = 9$; Fig. 1b–d; Supplementary Movies 1–3). In order to investigate the gain of the eye in relation to the head movement an accelerometer was fitted to the transparent tube ($N = 3$). The tube was then maneuvered in the three planes manually at velocities ranging between 60-200 deg/s while the lamprey eye was tracked. Eye and head positions were subsequently plotted (Fig. 1b–d) and compared in terms of their spatial and temporal alignment (Supplementary Fig. 1a). The dynamic gain was calculated by comparing eye-head velocities (see Methods). The gain was found to be $0.77 \pm 0.19$ for roll, $0.6 \pm 0.23$ for pitch, and $0.69 \pm 0.13$ for yaw. Position gain was retrieved by comparing the area-under-the-curve (AUC) between the eye and the head during the active movement. The gain values were $0.67 \pm 0.16$ for roll, $0.45 \pm 0.16$ for pitch, and $0.64 \pm 0.25$ for yaw.

In intact animals we could investigate the VOR behaviorally, but analyzing the contribution of visual and vestibular information, which entailed long experiments, was not possible to do in intact animals (see Methods). To study the interaction between visual and vestibular inputs, a setup was therefore developed that allows coordinated visual and vestibular stimuli via a tilting platform while performing electrophysiological recordings in an isolated preparation of the lamprey brain and rostral spinal cord with the eyes and vestibular organs (otic capsules) (Fig. 1e–g, Supplementary Movie 4). Visual stimuli consisted of horizontal bars moving in the vertical axis in opposite directions (i.e. when bars presented to the right eye move upwards, those for the left eye move downwards), reflecting visual inputs during a body-rotation. The oculo-vestibular-brain preparation was aligned with the rotating platform to produce vestibular stimulation in the roll plane.

To check the viability of the preparation, a video camera was attached in front, showing that VOR eye movements are evoked in response to roll stimulation (Supplementary Fig. 1b). To reliably quantify VOR throughout the study, we performed EMG recordings of the dorsal rectus extraocular muscle (DR; Fig. 1h). This muscle was consistently activated by downwards roll stimulation, in agreement with a compensatory eye movement in the opposite direction of the vestibular stimulation. Figure 1h (green trace) shows a representative response to a combined visual and vestibular stimulation (see also Supplementary Movie 5). Reliable responses were induced by isolated optokinetic (Fig. 1h, blue trace) or vestibular stimulation (Fig. 1h, orange trace). No activity was evoked when the platform came back to a horizontal position (Fig. 1h, green trace, red asterisk), indicating that the muscle activation corresponds to a VOR in this plane. To ensure that the EMG activity recorded in the extraocular muscles reliably corresponded to the eye movements, we performed electric stimulations at increasing intensities in the anterior octavomotor nucleus (AON; one of the vestibular nuclei involved in VOR, see below) while recording EMG activity in the dorsal rectus and simultaneously video recording eye movements (Supplementary Fig. 1c; $N = 2$). The EMG activity increased in parallel with the amplitude of the eye movements (Supplementary Fig. 1c–e; Supplementary Movie 6) and can thus be used to indirectly measure eye movements.

With this experimental platform we could monitor eye movements and record extracellular activity in different brain areas in response to a vestibular and/or visual stimulation.

### Optokinetic responses

Although visual stimuli (looming dots and bars presented to one eye) have been shown to evoke eye and orienting/evasive movements[24,25], an OKR had not been demonstrated in the lamprey. To test this, we applied a grid of bars moving in the three principal axes (roll, pitch, and yaw) at increasing speeds, and monitored EMG activity in the dorsal rectus (for roll and pitch stimulation) or the caudal rectus (for yaw stimulation).

There was a significant increase in the amplitude and spike numbers of the EMG activity in response to an increase of roll velocity ($n = 18$ from six lampreys). The graphs in Fig. 2a show that the responses in terms of EMG amplitudes (left) and spikes (right) increase in parallel with the speed of the stimulation (see representative traces in Fig. 2b). In Fig. 2e the combined data of six animals is shown. The same was true for the activity evoked in the pitch plane ($n = 15$ from five lampreys, Fig. 2f). Both EMG amplitudes (Fig. 2c, left) and number of spikes (Fig. 2c, right; see representative traces in Fig. 2d) increased with the stimulation speed. Surprisingly, no significant effect was found for stimulations in the yaw plane, which the lamprey will be subjected to during each swimming cycle (Fig. 1g, h; $N = 10$). Although visual responses could be evoked, they were very unreliable and no obvious increase in parallel with the visual stimulus velocity was observed (Fig. 2g, h). The same animals showed reliable OKR in both the roll and pitch planes (Fig. 2i, Supplementary Fig. 2a, b). Clear-cut optokinetic responses in the pitch and roll planes were thus established, indicative of compensatory eye movements in response to whole scene visual movements.

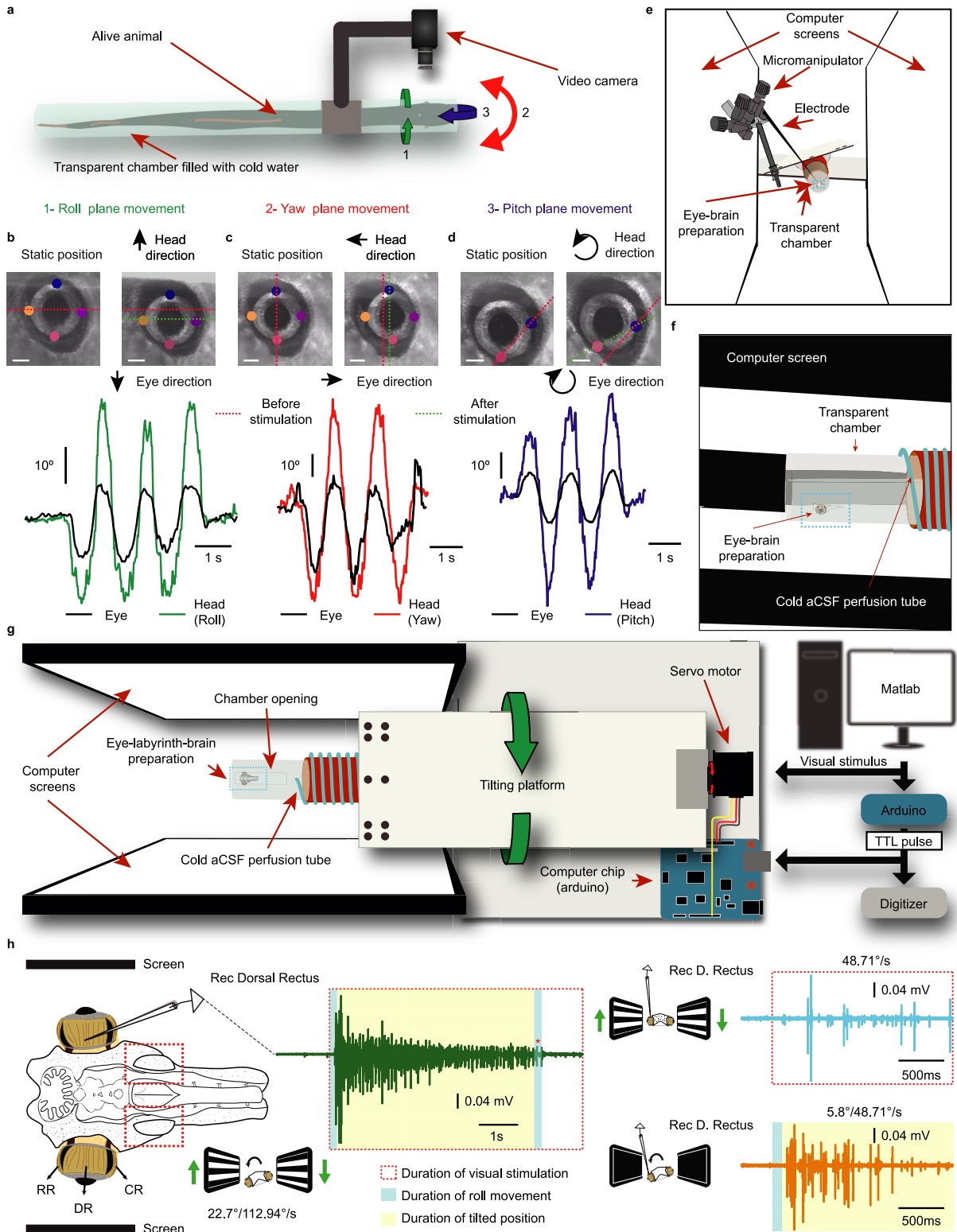

## Contribution of the visual and vestibular systems to the integrated motor response

Given that both visual and vestibular inputs contribute to gaze stabilization, the next question was how a combination of these two sensory modalities impacts on the compensatory eye movements. To test this, dorsal rectus activity was recorded in response to visual (VIS), vestibular (VES), and visuo-vestibular (VISVES) stimulation (Fig. 3a) under four distinct parameters in the roll plane: low amplitude−low speed (5.8°; 48.7°/s; n = 24 from eight lampreys), low amplitude−high speed (5.8°; 112.94°/s; n = 37 from 13 lampreys), high amplitude−low speed (22.7°; 48.7°/s; n = 24 from eight lampreys), and high amplitude−high speed (22.7°; 112.94°/s; n = 29 from ten lampreys). A continuous optokinetic stimulation was presented, to allow a better control over the evoked responses and a holistic evaluation of the visuo-vestibular integration. Analyzing both the initial dynamic visuo-vestibular stimulation and the subsequent responses of imposing only a dynamic visual component on a static vestibular signal allowed for temporal considerations and a nuanced perspective on gaze-stability. As

**Fig. 1 | A platform to perform electrophysiological recordings coordinated with visual and vestibular stimuli. a** Schematic showing the setup used to monitor VOR eye movements. **b–d** (Top) Representative frames before (left) and at the peak of stimulation (right) in response to rotations in the roll (**a**), yaw (**b**), and pitch (**c**) planes (Scale bars = 1 mm). The red dotted line indicates eye position before stimulation, and the green dotted line its position at the peak of vestibular stimulation. The colored dots in the images represent the labels used for the tracking system to calculate the eye trajectory. (Bottom) Traces representing the eye position (black) respect to the head position during several rotations in the roll (green), yaw (red), and pitch (blue) planes. **e–g** Schematic illustrations of the lamprey tilting platform. The system is controlled through Matlab, allowing for synchronized visual and vestibular stimulations through an Arduino controller board, which also engages a digitizer for electrophysiological recordings. The platform is moved with a servo-engine, controlled by another Arduino board, that rotates the transparent chamber containing the preparation together with the recording electrodes. A transparent chamber (containing an eye-labyrinth-brain preparation) connected to a tilting platform is placed between two screens as seen from the front (**e**) and side (**f**). An overall representation of the platform is shown in **g**. **h** Schematic showing the preparation used for recording activity in the eye muscles or different brain areas. The red squares indicate the location of the otic capsules, where the vestibular organs are located. A representative trace showing the activity evoked in response to a 22.7° tilting of the platform with angular speed of 112.94°/s, combined with coordinated visual stimulation, is shown to the left. To the right are similar recordings brought on by optokinetic stimulation of 48.71°/s (top), and vestibular stimulation in darkness of the same velocity at an amplitude of 5.8° (bottom). The blue area indicates the duration of the roll movement, yellow the duration of static tilt, and the dotted rectangle signifies ongoing optokinetic stimulation. Abbreviations: RR Rostral rectus, DR Dorsal Rectus, CR Caudal rectus, aCSF artificial cerebrospinal fluid.

differences in signal strength were observed between animals, the responses were normalized in each animal to the maximum intensities allowing us to then compare the results obtained in the different preparations and to obtain the ratio between sensory modalities.

**Low amplitude—low speed.** The individual VIS and VES responses were similar, but when applied together the VISVES activity was much larger, they thus reinforcing each other (Fig. 3b, f, j). Paired T-tests revealed a significant difference in the number of evoked spikes between VES and VISVES (Fig. 3b, j), but not for EMG amplitude (Fig. 3f).

**Low amplitude—high speed.** Visual and vestibular responses enhance each other under these conditions as well, as revealed in the significant differences between VES and VISVES for the number of evoked spikes (Fig. 3c, Supplementary Fig. 3a), and the maximum EMG amplitude increased significantly between VES and VISVES (Fig. 3g). VES number of spikes was significantly larger than the VIS one, indicating a larger contribution of the vestibular sensory information (Fig. 3c, Supplementary Fig. 3a).

**High amplitude—low speed.** The vestibular response was much larger than the VIS response and the addition of VIS did not add much to the VISVES response when comparing the number of spikes (Fig. 3d, Supplementary Fig. 3b), but when considering the EMG amplitude (Fig. 3h) VIS significantly added to VES.

**High amplitude—high speed.** In this case the vestibular response dominated and there was no significant addition by vision in the combined VISVES response (Fig. 3e, i, k).

**Temporal dynamics of visuo-vestibular interaction.** The above results show that joint visuo-vestibular stimulations enhance the responses, but the visual impact is more relevant at low amplitude movements. This, together with the fact that suprathreshold VIS responses are only evoked at the beginning of visual stimulation, suggests that its role is more important at the beginning of the compensatory eye movements. Vision contributes by increasing the number of evoked spikes (Fig. 3b–e), whereas integrated signal EMG amplitudes were significantly impacted only during the low amplitude−high speed and high amplitude−low speed conditions (Fig. 3f–i). This difference can be accounted for when considering the time at which the maximum EMG amplitude is reached for each sensory modality. For low amplitude conditions, the visual peak appeared later than the vestibular one (Fig. 4a–d), and this misalignment gives rise to a smaller impact of vision in terms of response EMG amplitude. The peak EMG amplitudes of VISVES were, as previously outlined, larger than VES for half of the conditions (Fig. 3f–i), although shifted towards the time of the visual peak for all conditions (Fig. 4a–d).

To capture the dynamics of visual contribution throughout the extent of the stimulation, we analyzed the number of spikes evoked under each of the three conditions (VIS, VES and VISVES) in windows of 100 ms, as shown in the heatmaps of Fig. 4e. A clear effect was that VISVES responses were prolonged as compared to only VES (see also traces in Fig. 3j, k and Supplementary Fig. 3a, b). There was an increase in the number of evoked spikes when comparing VISVES with VES trials even before the appearance of any activity evoked by only visual stimulation. Thus, a subthreshold increase in the excitability was evoked by VIS that was able to potentiate the vestibular responses (see also Fig. 3j, k and Supplementary Fig. 3a, b). Therefore, although the first visual spikes appear late compared to the vestibular responses (Fig. 4a–d), vision has an early impact when both sensory modalities are combined, and its contribution prolongs the evoked responses. The above results show that joint visuo-vestibular stimulations enhance the responses, but the visual impact is larger at low amplitude movements. This finding, together with that suprathreshold VIS responses are evoked only at the beginning of the visual stimulation, suggests that its role is particularly important at the beginning of the compensatory eye movements.

## Sources of visual and vestibular inputs to the oculomotor nucleus

Our results show that the interaction between visual and vestibular inputs determines eye movements underlying gaze stabilization. To identify the different brain regions that contribute to visual and/or vestibular information for gaze stabilization, we analyzed the inputs to the oculomotor nucleus (nIII), which innervates the dorsal rectus and is the last integration relay before movement initiation of the vestibular stimulation (connections are summarized in Supplementary Fig. 4a–h). Some of the pathways involved in VOR have been analyzed in lamprey larvae[17,20,31], but no data in adults are available.

Neurobiotin injections into the nIII (N = 4; Fig. 5a-inset) showed that the most rostral population projecting to this nucleus is in the ipsilateral thalamus (Fig. 5a). A few neurons were also found in the contralateral thalamus (not shown). Numerous neurons were retrogradely labeled in the ipsilateral pretectum (Fig. 5b), and some also contralaterally. Injections in pretectum (N = 2) showed numerous terminals in the oculomotor nucleus (not shown), confirming the direct pretectal projections to this nucleus. Projections were also observed from the nucleus of the medial longitudinal fasciculus (nMLF; Fig. 5c) and the SNc[32] (Fig. 5c). No retrogradely labeled neurons were observed in the optic tectum, suggesting that eye movements controlled from this area[21] are mediated via relay nuclei (presumed gaze centers), as in other vertebrates[33]. Rhombencephalic projections to the nIII arise from small neurons located in the area of the trochlear (nIV, Fig. 5d) and abducens (nVI, Fig. 5e) motor nuclei[31]. The most conspicuous labeling was found in the contralateral anterior

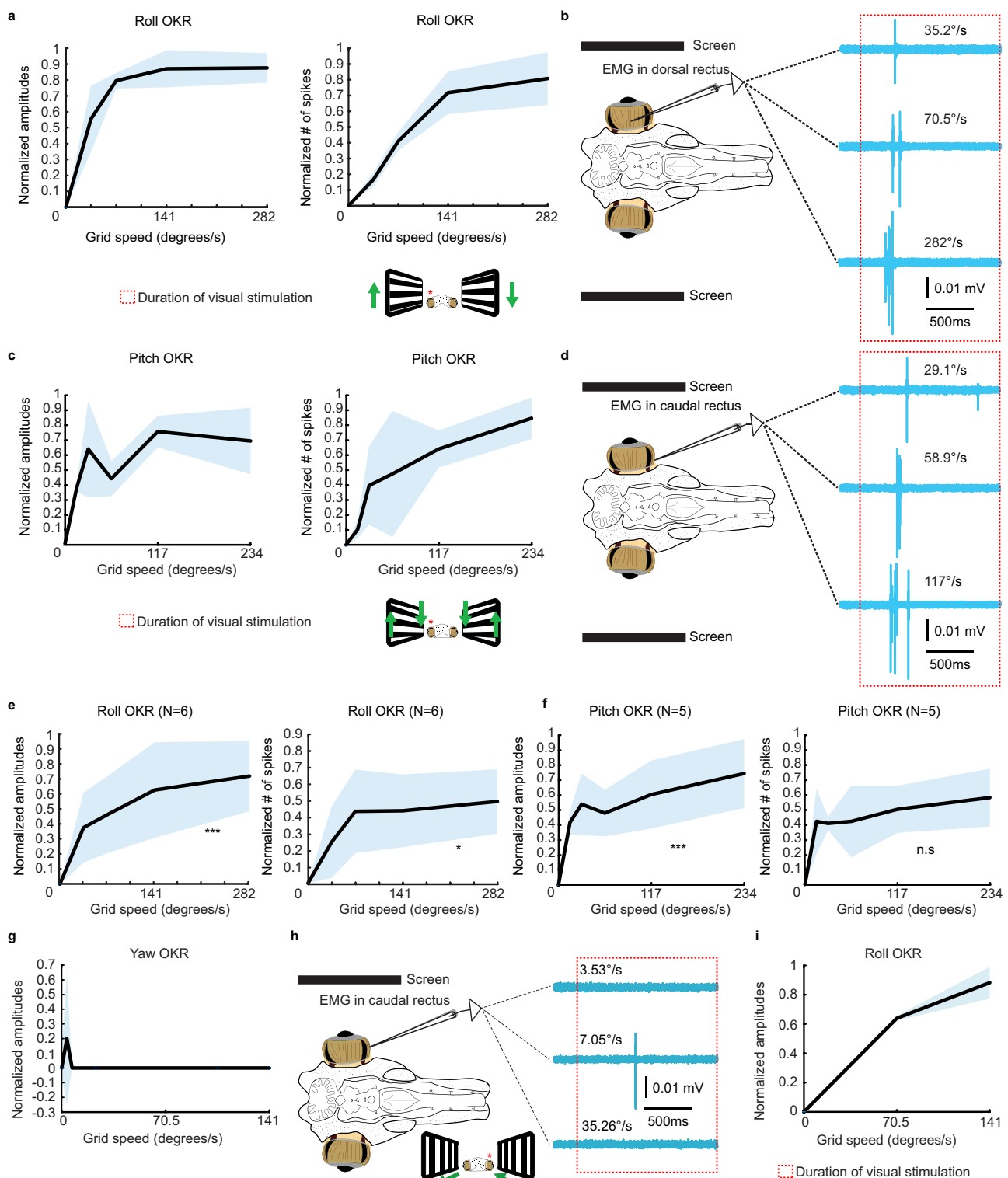

**Fig. 2 | Lamprey optokinetic responses. a** Graphical illustration of the normalized EMG amplitudes (left) and number of spikes (right) of the evoked EMG activity in the dorsal rectus of one animal during optokinetic stimulations in the roll plane across a range of velocities. **b** Representative traces at three different velocities during roll stimulations. The dotted rectangle indicates the duration of the optokinetic stimulation. **c** Graphs showing the EMG activity in the dorsal rectus of one animal during optokinetic stimulations in the pitch plane across a range of velocities. **d** Representative traces at three different velocities during pitch stimulations. **e**, **f** Mean EMG amplitudes and spikes reflecting the EMG activity in the dorsal rectus in response to optokinetic stimulation in the roll (**e**) and pitch (**f**) planes combining the data from six animals for roll, and five for pitch. A repeated measures ANOVA revealed significant effects of stimulation velocity on EMG amplitudes ($F[3, 51] = 7.858$, $p < 0.001$) and spikes ($F[3, 51] = 3.366$, $p = 0.026$) in the roll plane, as well as EMG amplitudes ($F[4, 56] = 5.057$, $p = 0.001$) in the pitch plane. **g** Graph showing that EMG activity does not increase in parallel to the speed of optokinetic stimulation in the yaw plane. Representative traces are shown in **h**. **i** Reliable optokinetic responses in the roll plane were however observed in animals lacking a yaw plane response. Holm corrections were carried out for all analyses. The shaded areas in the graphs denote error bands. Source data are provided as a Source Data file. Roll plane analysis was carried out on $n = 52$ independent samples from three animals, while pitch plane analysis was carried out on $n = 57$ independent samples from three animals.

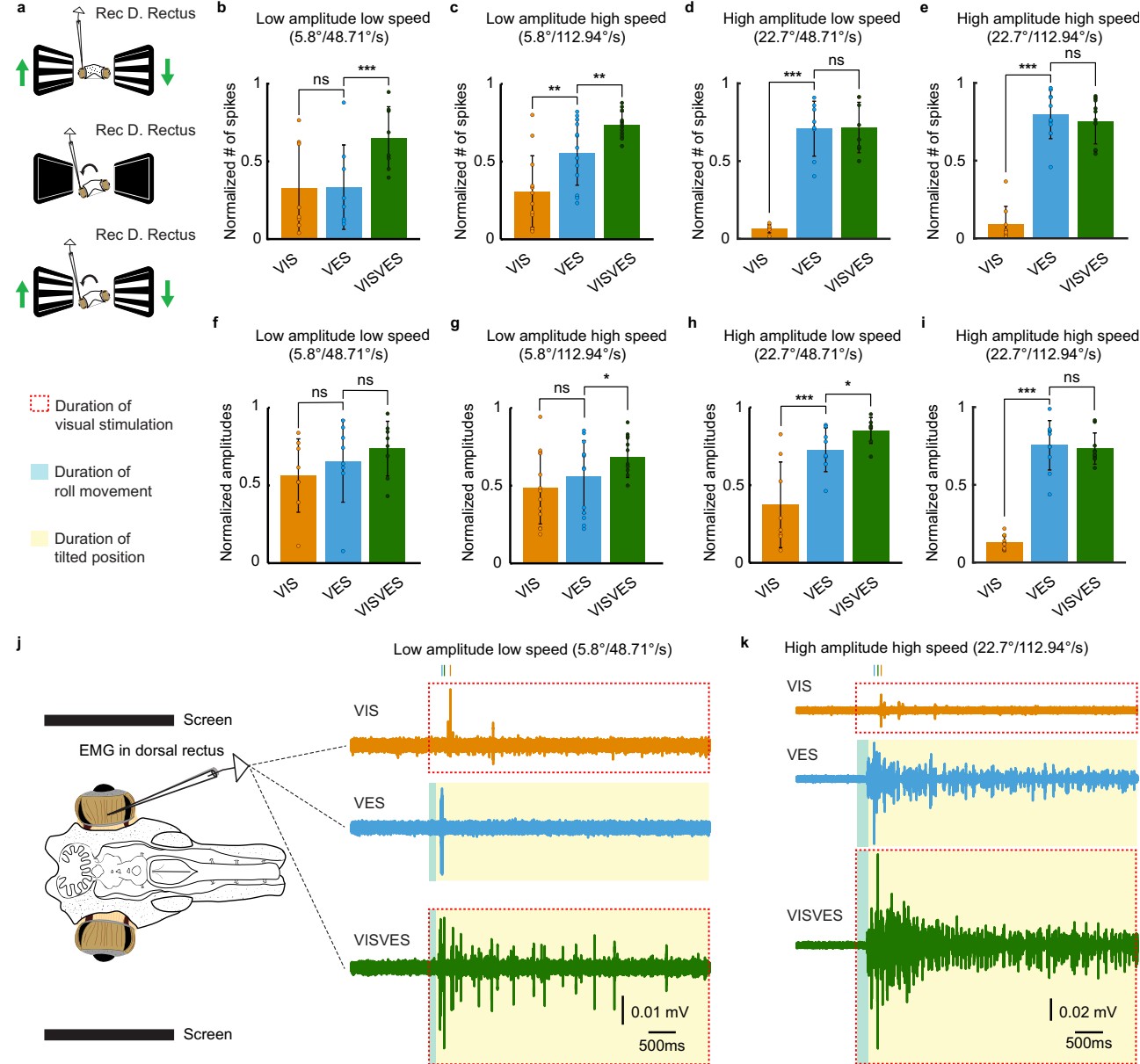

**Fig. 3 | Impact of visual and vestibular stimuli on eye movements. a** A schematic representation of the three experimental paradigms implemented: visual (top), vestibular (middle) and visuovestibular (bottom). Arrows indicate direction of stimuli movement for both visual (green) and vestibular stimulation (black). A recording electrode was placed in the right dorsal rectus muscle. **b–e** Graphs showing EMG activity in terms of spikes number in response to visual (VIS), vestibular (VES), and visuovestibular (VISVES) stimulations during four different stimulation protocols in terms of amplitude and velocity. Amplitude refers to the peak tilting angle of the platform, and speed to the motion velocity of the platform or the visual stimulus. Significance between modalities is represented by stars in each graph and was retrieved through paired *T*-tests: Low amplitude low speed (VES to VISVES t(23) = −4.802, *p* < 0.001; *n* = 24 recordings from eight animals), Low amplitude high speed (VIS to VES t(36) = −3.346, *p* = 0.002 *p* = 0.002 and VES to VISVES t(38) = −2.930, *p* = 0.006 *p* = 0.006; *n* = 37 recordings from 13 animals),

High amplitude low speed (VIS to VES t(22) = −11.559, *p* < 0.001, *p* < 0.001; *n* = 24 recordings from eight animals), High amplitude high speed (VIS to VES t(27) = −13.808, *p* < 0.001; *n* = 29 recordings from ten animals). **f–i** Graphs showing EMG activity in terms of maximum amplitudes in response to VIS, VES and VISVES stimulations during four different stimulation protocols ((**g**) VES to VISVES t(38) = −2.475, *p* = 0.018, *n* = 37 recordings from 13 animals; (**h**) VIS to VES t(23) = −6.315, *p* < 0.001, *n* = 24 recordings from eight animals; VES to VISVES t(22) = −2.614, *p* = 0.016, and (**i**) VIS to VES t(28) = −15.457, *p* < 0.001, *n* = 29 recordings from ten animals). **j, k** Representative responses for the lowest (**j**) and the highest intensity (**k**). The blue area indicates the duration of the roll movement, yellow the duration of static tilt, and the red dotted rectangle signifies ongoing optokinetic stimulation. All T-tests were two-tailed with no corrections. Throughout the figure, data are presented as mean values ± SD. Source data are provided as a Source Data file.

octavomotor nucleus, where large retrogradely labeled neurons were found (Fig. 5f; see also refs. 17 and 20). Coarse fibers arising from this nucleus could be observed crossing at different levels, as well as a few neurons on the ipsilateral side (not shown). These results show that nIII receives information from the pretectum and the vestibular nucleus, respectively.

## Pretectal and tectal influences on the optokinetic reflex

The anatomical results and data in other vertebrates suggested that pretectum is the primary contributor of visual information to OKR[8,9]. However, tectum plays a key role in visual processing[21–23,25] and therefore we aimed to identify which region is the primary contributor. We complemented the tracer injections with acute inactivation of

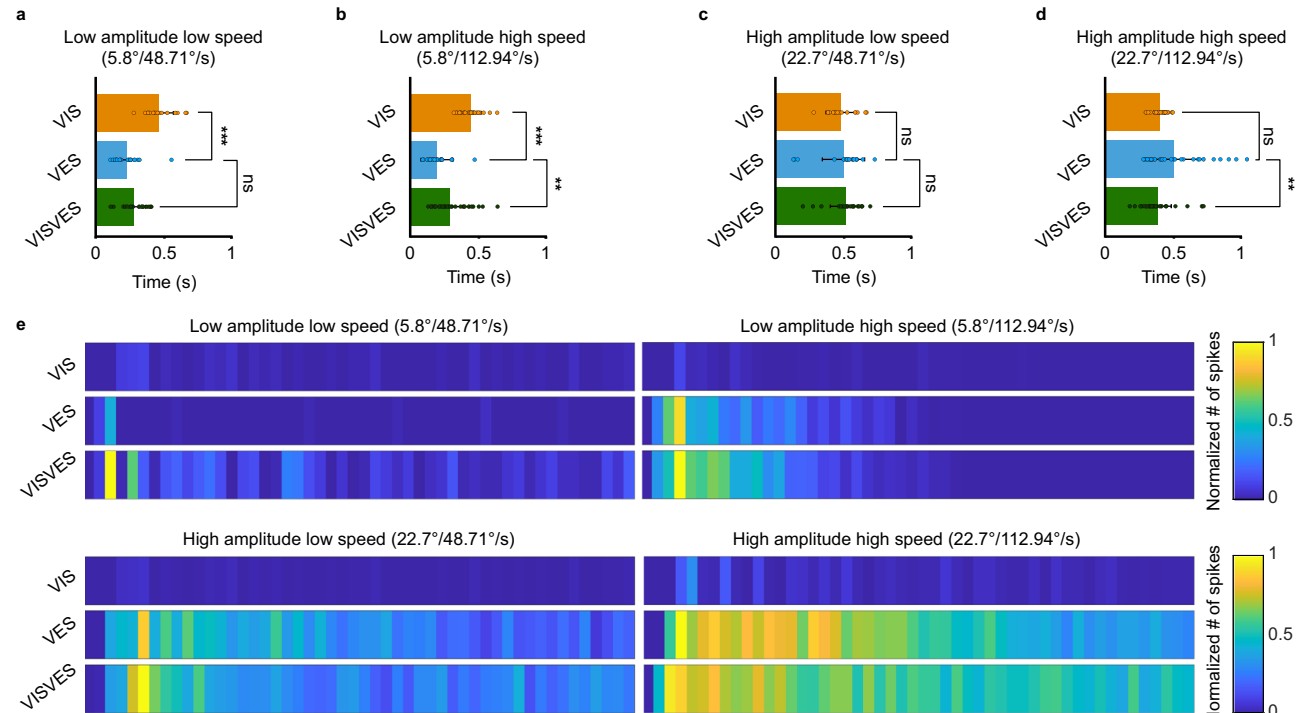

**Fig. 4 | Temporal dynamics of visuo-vestibular integration. a–d** Graphs showing the average time taken to reach the peak EMG amplitude for each modality during the different paradigms. Paired T-tests yielded significance results for: Low amplitude low speed (VIS to VES t(16) = 5.506, $p < 0.001$, $n = 17$ recordings from eight animals), Low amplitude high speed (VIS to VES t(23) = 10.871, $p < 0.001$, $n = 24$ recordings from eleven animals and VES to VISVES t(30) = −3.411, $p = 0.002$, $n = 31$ recordings from eleven animals), High amplitude high speed (VES to VISVES t(28) = 2.818, $p = 0.009$, $n = 29$ recordings from twelve animals). **e** Heat maps illustrating spiking activity during for four different paradigms for one representative animal. Values have been normalized to the highest density cluster, and dark blue signifies no spiking while yellow shows the greatest number of spikes. Each segment shows the spiking activity in a 100 ms windows, combining the averaged data from three different trials. All T-tests were two-tailed with no corrections. Throughout the figure, data are presented as mean values ± SD. Source data are provided as a Source Data file.

these two brain areas to see the impact on the dorsal rectus activity evoked by optokinetic stimulation.

When tectum was inactivated ($N = 5$) the OKR response remained intact (Fig. 6a red trace). In contrast, OKR responses were abolished when pretectum was inactivated either through lesion ($N = 3$; Fig. 6a blue trace) or an injection of the glutamate receptors antagonist kynurenic acid (KA; $N = 3$; Fig. 6b, red trace, responses recover after washout, green trace), even when tectum was intact. The plot in Fig. 6c shows that there is a significant reduction of OKR after KA injection in pretectum (t(6) = 2.633, $p = 0.036$), and that the OKR significantly recovers after washout (t(3) = −3.703, $p = 0.034$).

A homolog of the visual cortex is present in the lamprey pallium with vision represented retinotopically[26], and electric stimulation of this region generates eye movements[28]. No significant impact was observed on the visual responses evoked by optokinetic stimulation after lesioning the visual area, indicating that the pallium is not directly involved in generating OKR (Supplementary Fig. 4i; $n = 12$ from four lampreys).

We then analyzed the changes in visuo-vestibular integration after lesioning tectum, to see if visual information from pretectum still had an impact on vestibular responses. VISVES responses were still significantly larger than only VES (Fig. 6d, e), indicating that this enhancement is not dependent on tectum. The pretectum consequently appears key in relaying visual information to downstream subcortical structures allowing the visuo-vestibular integration for gaze-stabilization. Contrary to the pretectal lesioning, inactivation of tectum increased the eye muscle activity to visual stimulations (Fig. 6a, red trace). This was manifested in significantly greater EMG amplitudes during VISVES trials post-lesion (Fig. 6f), and that this peak was

reached earlier (reduced from 0.23 ± 0.05 s to 0.19 ± 0.02 s, $t$ [12] = 2.881; $p = 0.014$). Consequently, tectal inactivation enhanced the visuovestibular integration, yielding greater EMG amplitudes and shorter response times.

## The vestibular nuclei receive visual information

In other vertebrates, vision impacts the vestibular nuclei independently of projections to the oculomotor nuclei[34–36]. We investigated whether this applies also in the lamprey. The vestibular nuclei homolog is divided into anterior (AON), intermediate (ION), and posterior octavomotor nuclei (PON). The vestibular inputs to the oculomotor nucleus innervating the dorsal rectus come from the AON and ION (present work, 20). We therefore performed extracellular recordings in these two nuclei (Fig. 7a; $n = 39$ from 13 lampreys). In both nuclei visual field-rotation stimulation evoked responses (Fig. 7a; black trace). These responses were abolished by inactivation of pretectum ($n = 9$ from three lampreys; Supplementary Fig. 5a), but not by that of tectum ($n = 9$ from three lampreys; Fig. 7a; red trace) and pallium ($n = 15$ from five lampreys; not shown). This indicates that only pretectum provides optokinetic information to the vestibular nuclei.

We then performed Neurobiotin injections into the AON ($N = 4$) and ION ($N = 4$), to map the origin of the visual information reaching these vestibular nuclei. Both AON and ION receive the same inputs, and therefore the described results apply to both nuclei. Injections in the AON and ION (Fig. 7b, inset) confirmed projections to the oculomotor nucleus (Fig. 7b), and both showed ipsilateral projections terminating in the nMLF (Fig. 7b, e). The most rostral population targeting these nuclei was found in the same thalamic area

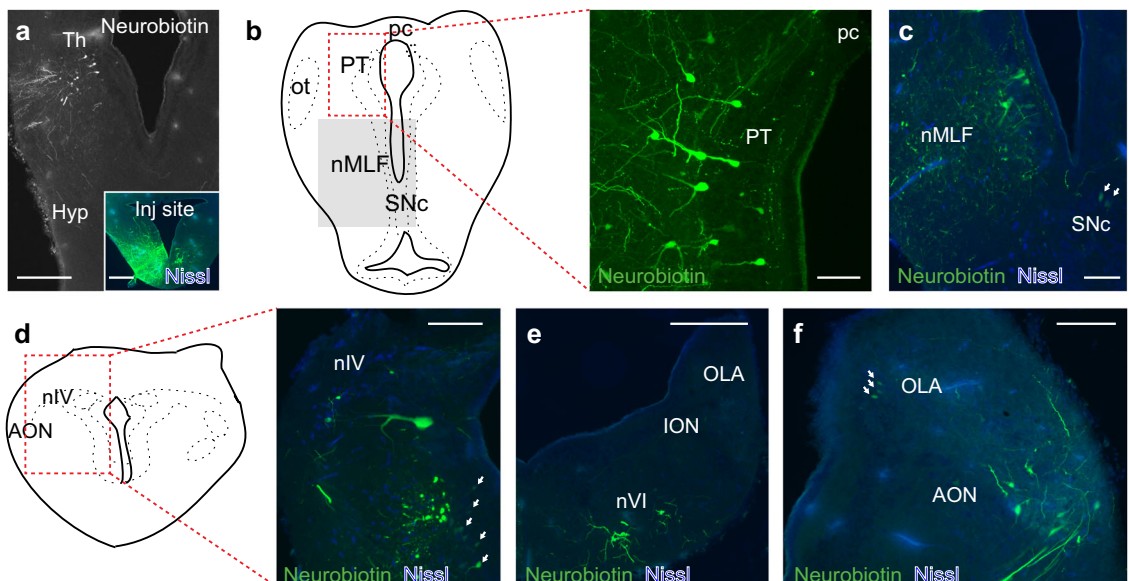

**Fig. 5 | Visual and vestibular sources to the oculomotor nucleus. a** Neurobiotin was injected in the oculomotor nucleus (inset), which showed ipsilateral projections from thalamus. **b** Numerous retrogradely labeled cells were found in pretectum forming a population projecting to the ipsilateral oculomotor nucleus. **c** Retrogradely neurons in the region of the nMLF. Some retrogradely labeled neurons were also observed in the SNc (arrows). **d, e** Projections were also found from the two other cranial nerve nuclei responsible for extraocular muscle innervation, illustrated here by a cell population in the trochlear nucleus (**d**) and contralateral abducens nucleus (**e**). Retrogradely labeled neurons were also found in the ventral part of the isthmic region, nearby the thick axons labeled from the AON. (**d** arrows). **f** a strong projection was confirmed[17,20] from the AON, and retrogradely labeled neurons were also found in dorsal aspects of the OLA. Abbreviations: Th Thalamus, Hyp Hypothalamus, pc posterior commissure, PT Pretectum, ot Optic Tract, nMLF Nucleus of the Medial Longitudinal Fasciculus, SNc, Substantia Nigra *Pars Compacta*, nIV Trochlear Motor Nucleus, AON Anterior Octavomotor Nucleus, OLA Octavolateral Area, ION Intermediate Octavomotor Nucleus, nVI Abducens Motor Nucleus. Scale bar = 250 μm in **a** (and inset) and **e**; 100 μm in **b** and **c**; 150 μm in **d** and **f**.

that sends projections to the oculomotor nucleus (Fig. 7c; see above), suggesting that this region is involved in gaze stabilization. No retrogradely neurons were observed in the tectum. Labeling was however found in pretectum, as expected based on the electrophysiological experiments (see above) which corroborate the notion that the visual information originates in pretectum (Fig. 7d). Although a few neurons were observed in medial aspects of the pretectum, most retrogradely labeled neurons were located close to the ipsilateral optic tract where their dendrites extended (Fig. 7d). Numerous retrogradely labeled neurons were observed in the contralateral AON (and ION for injections in this nucleus), indicating that vestibular nuclei on both sides influence each other (Fig. 7f).

To confirm that visual information impacts the vestibular, we used a preparation exposing the AON neurons for patch-clamp recordings while maintaining the pretectum (Fig. 7g) and the optic tract. Then injections of dextran-rhodamine were made in the AON tract to retrogradely label and identify the AON neurons for patch-clamp recordings (Fig. 7g; n = 8). Neurons showed excitatory responses to pretectal stimulation (10 Hz; Fig. 7h, red trace; n = 7), with all evoked EPSPs having similar amplitudes (slightly depressing; Fig. 7j). No inhibition was revealed when neurons were held at more depolarized levels (Fig. 7h, blue trace). The lack of inhibitory responses was confirmed intracellularly blocking sodium channels with QX314 (n = 2) that allowed depolarization to −20mV (Supplementary Fig. 5b). Neurons projecting to the oculomotor nucleus showed little adaptation (Fig. 7i) and afterhyperpolarization (Fig. 7i, blue trace). Altogether, these results show that visuovestibular integration takes place at the level of the vestibular nuclei, and that visual information arises in the pretectum. When the AON neurons were not prelabelled, EPSPs were evoked only in some neurons in the region of the AON (4/23 neurons recorded), and IPSPs only in one neuron (not shown) suggesting that only a proportion of the neurons within the AON region interacts with pretectum.

## Vestibular nystagmus is also present in lampreys

As described above, lampreys possess a well-developed visual system with eye movements that can be evoked with non-optokinetic visual stimuli[24]. This suggests that they may be able to perform goal-oriented saccades, and, if true, not only the slow but also the quick resetting phase of nystagmus could be evoked with vestibular stimulation[37]. To analyze nystagmus, we developed a platform that allows a full rotation of intact lampreys while video tracking eye movements (Fig. 8a). Rotations evoked compensatory eye movements (VOR) but also resetting movements in the opposite direction (Fig. 8b–c; Supplementary Movie 7). To investigate whether the lamprey VOR features nystagmus eye movements, we applied 180° rotations at three different velocities (27.4, 68.5, and 137 °/s), and measured the duration of each eye movement episode (N = 3). In agreement with nystagmic movements, the duration and speed of the slow-compensatory eye movements changed with the rotation velocity (Fig. 8d), while the duration of the resetting phase yielded very similar durations (0.024 ± 0.009 s) across different stimulation velocities in a fashion typical of a quick-phase nystagmic eye movement (Fig. 8d), showing that movements of saccadic nature are present in the lamprey. The analysis of the dynamics of these resetting eye-movements revealed that their velocity was 77.06 ± 30.30 deg/s and the amplitude 1.72 ± 0.73°, with no significant differences when comparing quick phases evoked at different stimulation velocities.

## Locomotive influences on compensatory movements by corollary discharges

The lack of reliable OKR responses in the yaw plane raised the question of whether other mechanisms apart from VOR compensate for head movements in this plane. First, we tested if the head is stabilized while swimming by monitoring freely moving animals (Supplementary Movie 8), but head movements accompanied swimming undulations, indicating that clamping of the visual scene is not achieved by head stabilization. Another possibility established in other vertebrates[38–40] is

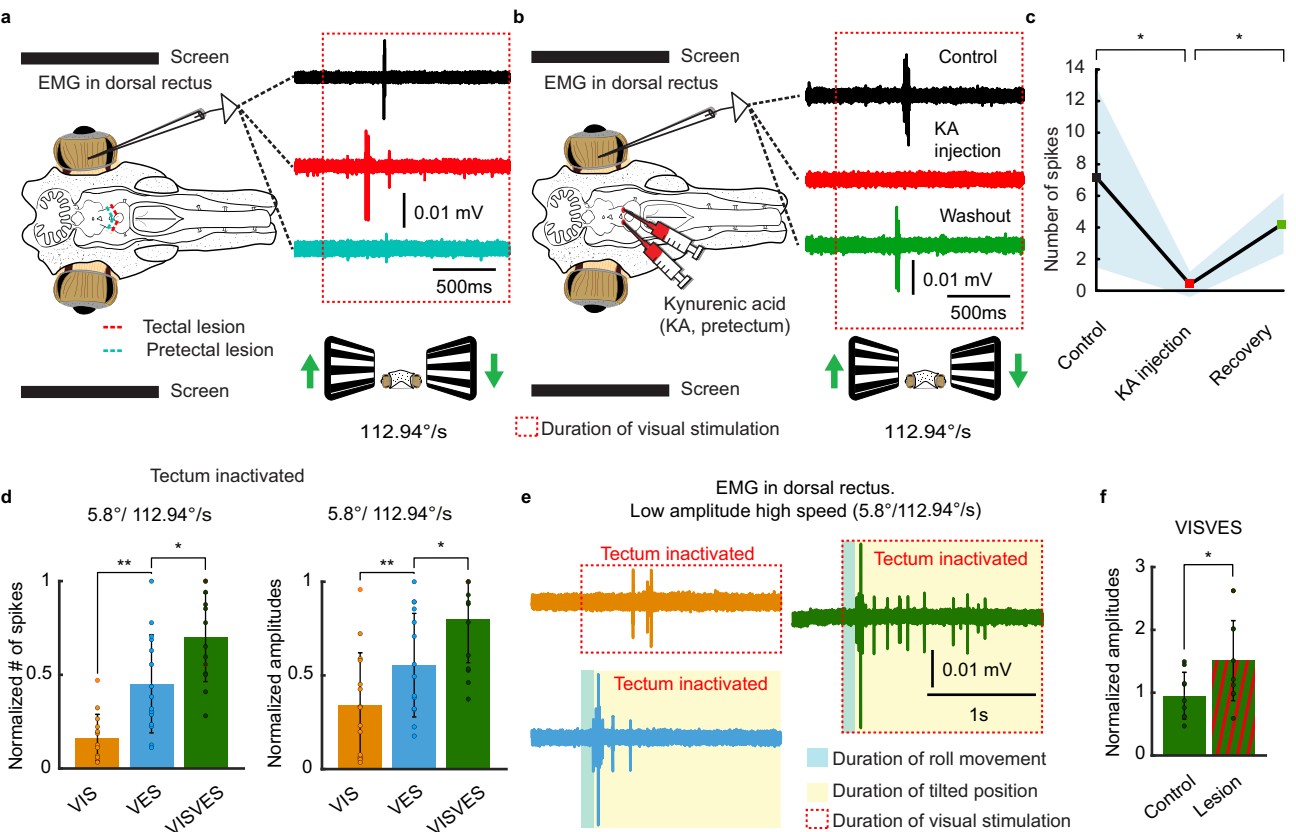

**Fig. 6 | Optokinetic responses are mediated by pretectum and downregulated by tectum. a** EMG responses in the dorsal rectus to optokinetic stimulation in an intact brain (black trace) and after precise inactivation of the visual input to tectum (red trace) and pretectum (blue trace). **b** EMG responses in the dorsal rectus to a visual stimulus in an intact brain (black trace) and after precise pharmacological inactivation with kynurenic acid of pretectum leading to the abolishment of the optokinetic reflex (red trace), which returned after a washout (green trace). The red dotted rectangle indicates the duration of the visual stimulation. **c** graph showing the significant reduction in dorsal rectus activity in response to optokinetic stimulation after pretectal inactivation with kynurenic acid (KA; t(6) = 2.633, $p = 0.039$, $n = 7$ recordings from three animals), and the significant recovery after washout (t(3) = −3.703, $p = 0.034$, $n = 4$ recordings from three animals). The shaded area denotes error bands. **d** Graphs showing the normalized EMG amplitudes (left) and spikes (right) to visual (VIS), vestibular (VES), and visuovestibular (VISVES)

stimulations in the roll plane after tectal inactivation. Paired T-tests revealed significant differences in the number of evoked spikes between VIS to VES (t(13) = −3.277, $p = 0.006$, $n = 14$ recordings from five animals) and VES to VISVES (t(14) = −2.740, $p = 0.016$, $n = 15$ recordings from 5 animals), as well as in maximum EMG amplitudes (VIS to VES t(14) = −3.717, $p = 0.002$; and VES to VISVES t(14) = −2.917, $p = 0.011$; $n = 15$ recordings from five animals for both). **e** Lamprey eye movement responses to VIS, VES, and VISVES stimulations in the roll plane during tectal inactivation as reflected by representative EMG recordings in the dorsal rectus. **f** VISVES responses were significantly larger after tectal inactivation for both amplitudes (t(14) = −3.005, $p = 0.009$, $n = 15$ recordings from five animals) and spikes (t(13) = −2.469, $p = 0.028$, $n = 14$ recordings from five animals). All T-tests were t-tailed with no corrections. Data are presented as mean values ± SD. Source data are provided as a Source Data file.

that locomotor networks through an efference copy drive compensatory eye movements in the yaw plane. To test this in the lamprey, we used a semi-intact preparation[21] (Fig. 9a). We first immobilized the head of the preparation to monitor whether compensatory eye movements occurred by using a video camera placed on top to monitor tail and eye movements (Fig. 9a). In one out of 6 animals coordinated eye movements were generated coordinated with swimming frequency (Fig. 9b–f; Supplementary Movie 9). Although reduced, these movements persisted when the labyrinths were removed and the optic nerves sectioned (Fig. 9d, g), indicating that they arise from corollary discharges.

Given that clear compensatory eye movements coordinated with locomotion were observed, albeit only in one animal, we investigated whether a small activation of the extraocular eye muscles that did not result in perceivable eye movements could be recorded via EMGs. As swimming corollary discharges have been shown to arise at the spinal cord level[38], we performed EMG recordings in the rostral and caudal rectus muscles (which should be activated in the yaw plane compensatory movements) while simultaneously recording in a pair of ventral roots in the spinal cord (Supplementary Fig. 6a; $N = 4$). Recordings

were carried out in a split chamber, and D-glutamate (750 μM) was applied to the spinal cord without affecting the brain (Fig. 6a). D-Glutamate evoked fictive locomotion[41], as recorded in the ventral roots (Supplementary Fig. 6b, top traces). However, no compensatory responses were evoked in the eye muscles (Supplementary Fig. 6b, bottom traces). Altogether, these results show that corollary discharges can generate compensatory eye movements, as clearly observed only in one animal, although its contribution is in most cases subthreshold or absent, and its origin remains to be determined.

## Discussion

Our findings show that not only VOR, but also OKR and saccadic eye movements in the form of nystagmus are present in lampreys. Using a platform that allows combined visuovestibular stimulation with electrophysiological recordings, we show that the impact of visuovestibular integration on eye movements is similar to mammals, demonstrating that neither cortical nor cerebellar influences are required for the multisensory integration underlying enhanced eye movements. We show that gaze-stabilizing eye movements, including nystagmus, rely on a few fundamental subcortical structures (Fig. 10a).

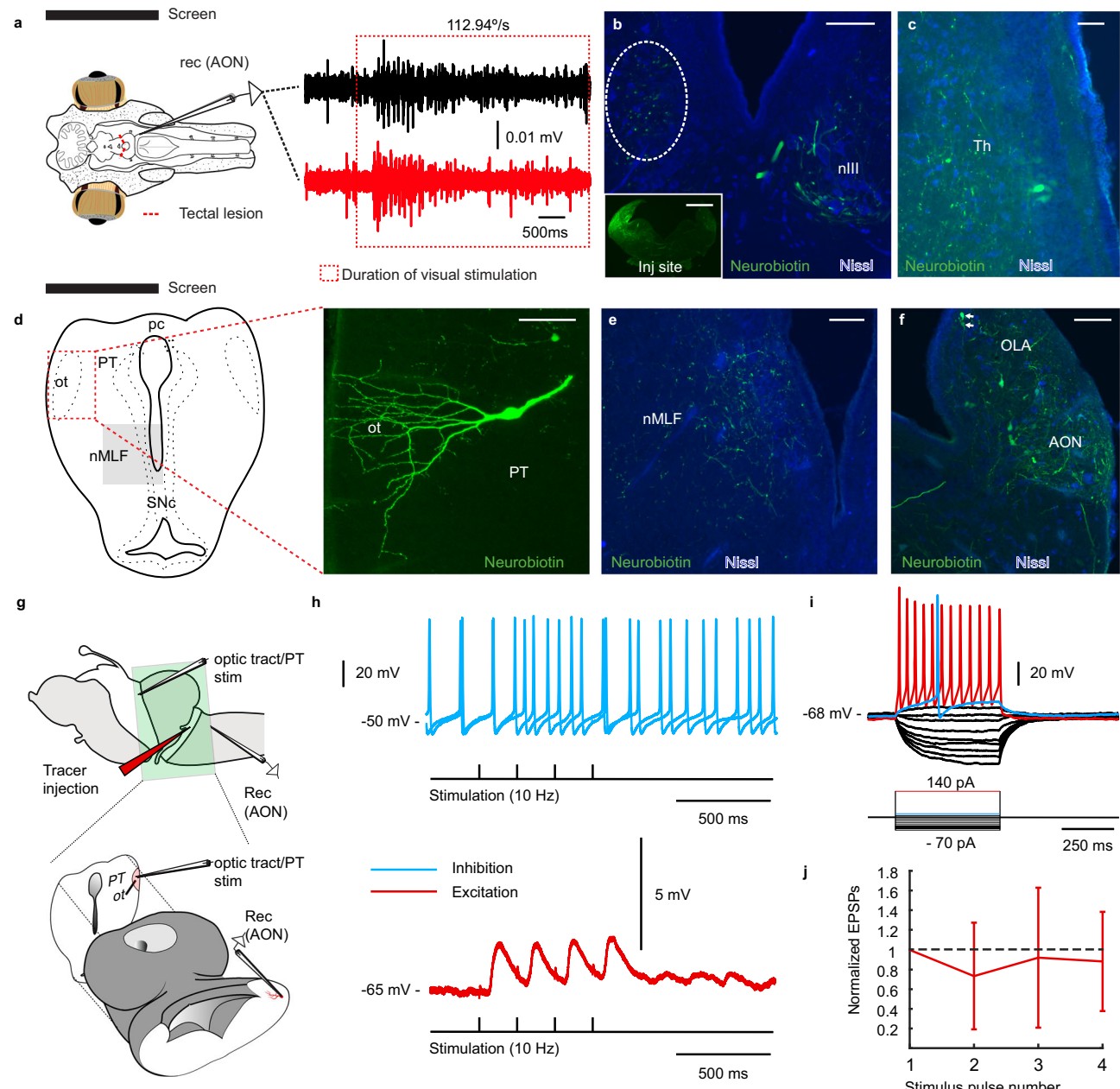

**Fig. 7 | Visual inputs to the vestibular nucleus. a** (left) Recordings in the AON in response to optokinetic stimulation during normal conditions (black trace) and after tectal lesioning (red trace). The red dotted rectangle indicates the duration of the visual stimulation. **b** Anterogradely labeled fibers terminating in the contralateral nIII, after a neurobiotin injection into AON (inset). Labeled fibers also course more caudally towards the nMLF (dashed oval; see **e**). **c** Retrogradely labeled cells were found in the ipsilateral thalamus **d** Retrogradely labeled cells can be seen in pretectum (right) with dendrites reaching into the optic tract. **e** Terminals in the ipsilateral nMLF. **f** Contralateral projections at the level of the injection site, revealing significant cross-talk between vestibular areas. **g** A schematic showing the thick section of the lamprey brain used for intracellular patch-clamp recordings, maintaining the pretectum and exposing AON for whole-cell recordings. A tracer injection was previously made in the tract from the AON to the nIII, at the level of the isthmus, allowing for visualizing projection neurons in the AON for whole-cell recordings. **h** Excitatory (bottom, red trace) responses of a

representative cell in a prelabelled AON neuron during a four pulses stimulation (10 Hz). No cessation of spikes, indicative of inhibitory inputs, was observed when neurons where depolarized (top, blue trace). **i** Voltage responses to hyperpolarizing and depolarizing 500 ms current steps of 10 pA per step, elicited from rest at −68 mV, showing threshold (blue trace) and suprathreshold response (red trace). **j** Quantification of excitatory postsynaptic potential (EPSP) amplitudes in AON cells projecting to nIII evoked by sustained stimulation (10 pulses at 10 Hz) of the pretectum/optic tract recorded in current-clamp mode. Values are normalized to the first EPSP. Data are presented as mean values ± SD. Abbreviations: AON Anterior Octavomotor Nucleus, nIII Oculomotor nucleus, Th Thalamus, pc posterior commissure, PT Pretectum, ot Optic Tract, nMLF Nucleus of the Medial Longitudinal Fasciculus, SNc Substantia Nigra *Pars Compacta*, OLA Octavolateral Area. Scale bar = 150 μm in **b**; 250 μm in b-inset; 100 μm in **c**–**f**. Source data are provided as a Source Data file.

Previous studies have stipulated that the OKR emerged first in chondrichthyes[1]; our findings push the onset of visually guided gaze-stabilization further back in an updated phylogenetic tree of vertebrate eye movements (Fig. 10b). We also show that the eye-movements clearly

compensate for the head rotation, and that locomotion-evoked gaze-stability is present as in a number of vertebrate species. Altogether, this study introduces a number of key reference points for the evolution of eye movement control and elaborates on its fundamental mechanisms.

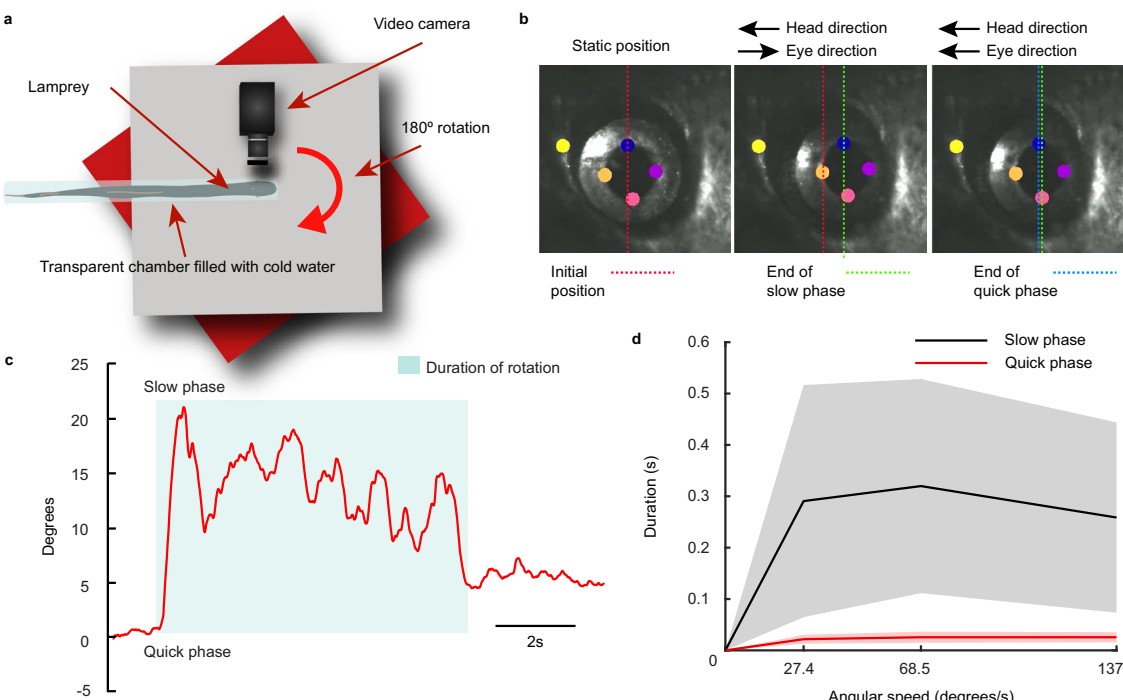

**Fig. 8 | VOR nystagmus. a** Our in-lab built platform allowed for controlled rotations of intact lampreys in the yaw plane. Animals were encased in a transparent plastic tube filled with cold fresh water. The diameter of the cylinder allowed for breathing while limiting the animal's freedom of movement. A camera was attached to the platform so that one eye could be filmed during the stimulation, which consisted in 180° rotations at different speeds. **b** Lamprey eye positions were quantified using DeepLabCut, which allowed for identifying movements of the pupil (four labels were used to average, indicated in orange, blue, purple and pink) in reference to the head (yellow). The red dotted line represents eye position at the start of the yaw stimulation (static position), while the green line indicates the end position of the slow (compensatory) phase result of a movement in opposite direction to that of the head. The blue line indicates the end position of the eye after the following quick phase (resetting) eye movement in the same direction as the head movement. **c** Using the eye-tracker previously outlined, the eye position could be translated into angular degrees over time, revealing a clear sawtooth-pattern indicating a VOR with nystagmus. **d** The durations of eye movement slow-phases (black) and quick-phases (red) were plotted for the duration of the rotational yaw movement. Durations were calculated based on frame-by-frame analysis of the video recording. (27.4, 68.5, 137°/s). As indicated in the graph, the quick-phase eye movements (red) were consistently of the same general duration across trials as compared to slow-phases (black), which also showed higher variability among different trials at the same speed, as reflected in their larger standard deviations as compared to quick-phase eye movements. Data are presented as mean values ± SD. The shaded areas denote error bands. Source data are provided as a Source Data file.

The VOR in lamprey reliably compensates for head movements in all three planes, with eye movement amplitudes reflecting the dynamic properties of the lamprey oculomotor system, its sensorimotor integration, and the presence of gaze-stabilization in early vertebrates. The dynamic gain was recorded to be around 0.7 while the position gain approximated 0.6. These values are largely in line with previous studies on teleosts[42], while further trials are needed in order to further explore the lamprey VOR gain. VOR can also be observed in ex vivo preparations, allowing for visuovestibular experiments in a highly controlled fashion. Here we confirm that the basic circuit underlying VOR via disynaptic pathways had evolved already in lampreys and thereafter maintained throughout vertebrate evolution albeit with species-specific modifications[10,17,20], and that, apart from nIII, visuovestibular integration also takes place in the vestibular nuclei and nMLF. The available data suggest that the disynaptic arch underlying the VOR appeared in early vertebrates supported by the anterior and posterior semicircular canals and that it has been largely maintained in all vertebrate groups[10]. The appearance of a lateral horizontal canal in gnathostomes, which is Otx dependant[17,43], gave rise to some circuit arrangements[17]. Namely, the six extraocular muscles exhibited by the lamprey are thought to be homologous of those of bony fishes and tetrapods[13,17], but with some differences in their motor nuclei innervation as compared to gnathostomes. Three muscles are innervated by the oculomotor nucleus (nIII), one is innervated by the trochlear nucleus (nIV), and two by the abducens (nVI). In elasmobranchs four muscles are innervated by the nIII, while one is innervated by the nIV

and one by nVI. In bony fish and tetrapods four muscles are innervated by the nIII, one by the nIV, and at least one by nVI[17,44]. However, the same computational scheme, also present in lampreys, was maintained. Interestingly, lampreys exhibit an overall pattern of second-order vestibular projections very similar to mammals[17]. The vestibular system thus shows highly conserved elements as well as variations related to specific functional variations[10,45].

When triggering the OKR we ensured that the visual stimulation covered the entire visual field but noted that responses disappeared when the same stimulus was applied at a greater distance, further supporting the optokinetic nature. OKR was reliably seen in the roll and pitch planes, with a sharp and immediate increase in both spiking and EMG amplitude as the grid speed increased before levelling out, in agreement with increased velocities making the visual clamping more difficult[46]. Our experiments show that the OKR relies on pretectal activity as in sharks, bony fish, amphibians, reptiles, birds and mammals[9]. Accordingly, as in other vertebrates[47], tectal inactivation does not abolish OKR.

The clear interconnectivity between pretectal, vestibular, and oculomotor nuclei also supports the notion that both OKR and VOR are of subcortical origins, highlighting a principal role of subcortical mechanisms, which has likely been maintained in mammals due to the phylogenetic preservation of the visual system[15,23,25,26]. The absence of a functional cerebellum in lampreys shows that cerebellar pathways are not needed for the OKR. Pretectal information also reaches the vestibular nuclei in the lamprey, indicating that the pretecto-vestibular

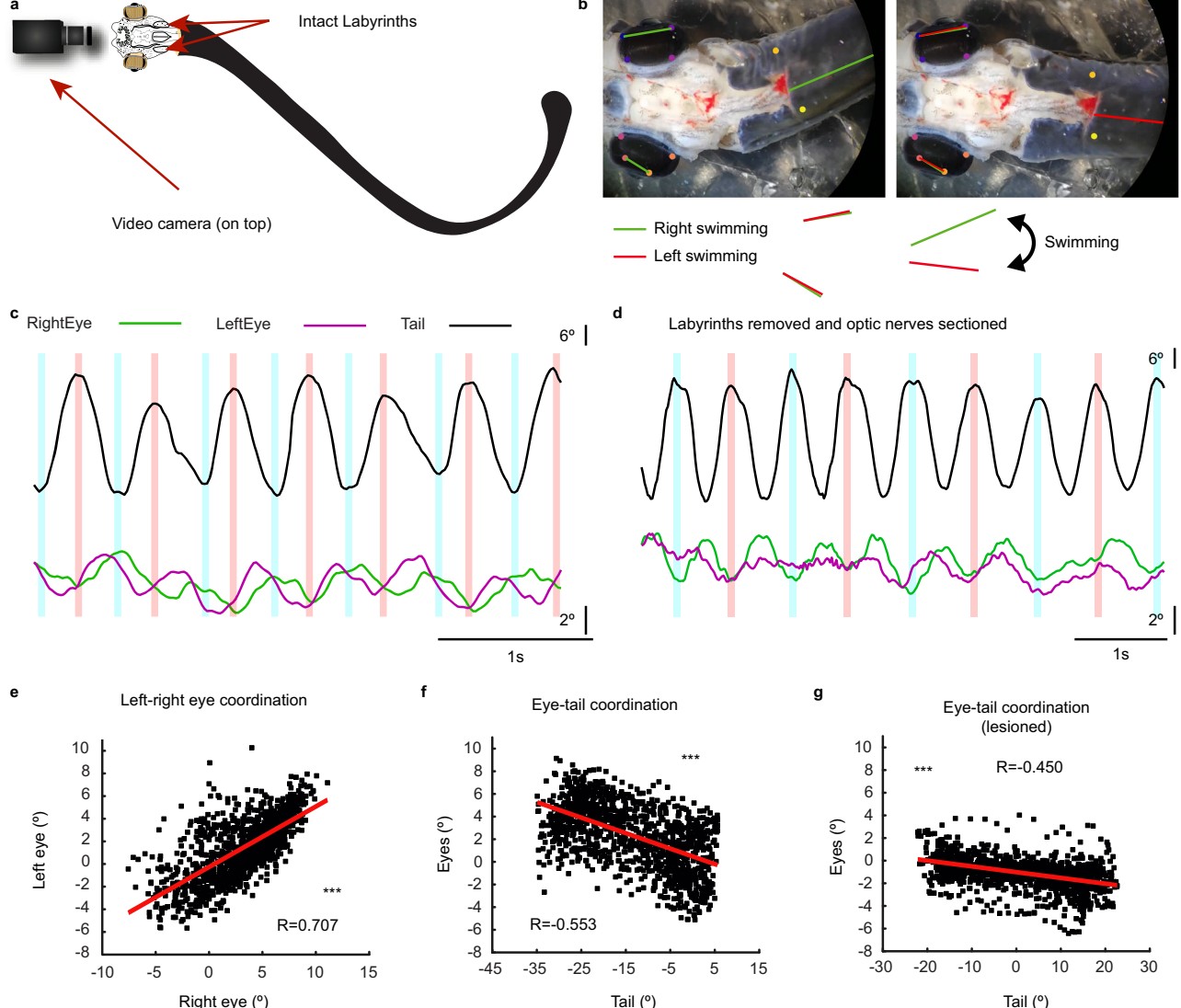

**Fig. 9 | Locomotion evoked eye movements. a** Schematic showing the semi-intact lamprey preparation used to monitor body and eye movements. The rostral segment up to the spinal cord is dissected according to the same principles as the ex vivo preparation, exposing the brain and the eyes. The remainder of the body and tail were kept intact in order to allow locomotion. A video camera was placed coupled to a microscope to film the preparation from above. **b** Either spontaneous or tactilely induced locomotion (by gently pinching the tail) was recorded, and eye and body movements were analyzed over time. Eye movements synchronized with swimming activity were observed. Images show two different positions of the eyes and the tail during a swimming episode. **c** Trajectories of the tail (black) and eyes (green, right eye; purple, left eye) showing that coordinated movements of both eyes occur together with tail movements. **d** Although these movements were less consistent, they were preserved after visual and vestibular inactivation. **e** Graph showing the positive correlation (Pearson's correlation analysis, r (1166) = 0.707, p < 0.001) between the right and left eyes, indicating their synchronization. **f, g** Graphs showing that tail and eye movements are correlated both before (**f**; Pearson's correlation analysis, r (1160) = −0.553, p < 0.001), and after (**g**) visuovestibular inactivation (r (1582) = −0.450, p < 0.001), indicating that the observed coupled eye movements are generated by locomotion corollary discharges. Source data are provided as a Source Data file.

pathway was already present in early vertebrates[48]. Lamprey responses to rotational visual motion imply pretectal binocularity given the lateral position of the eyes[49], although the underlying mechanisms of whole-field visual processing remain to be investigated[50].

Surprisingly, no OKR was observed in the yaw plane, in which most movement occurs in each swim-cycle. However, although our results show that corollary discharges alone in most cases do not generate compensatory eye movements, they seem to provide a sub-threshold contribution that would compensate for the lack of OKR in this plane (Fig. 9).

OKR and VOR enhance one another, but at higher velocities the relative contribution of the vestibular response increases. The vestibular stimulation noticeably produced a strong, long lasting eye movement as if the eye is compensating for the head displacement. It

is therefore clear that additional visual input decidedly increases eye movement gain in lamprey as well as humans, providing further evidence that the underlying mechanisms are conserved[6,7,51]. Visual inputs from pretectum also potentiate vestibular responses controlling body posture in lampreys[52,53], suggesting a key role of pretectum contributing to both gaze and posture stabilizing responses[54].

Lampreys also show nystagmus, characterized as quick-phased, ballistic eye movements with fixed durations in the opposite direction of the VOR slow-phase[55]. The lamprey has thus far been considered to not display nystagmus[16], but our results show a clear quick-phase, resetting eye movements as part of the VOR, with similar durations independently of stimulation velocity or eye movement amplitude. Eye movements occurring during rotations produce intermittent quick- and slow-phases making up the sawtooth pattern characterizing

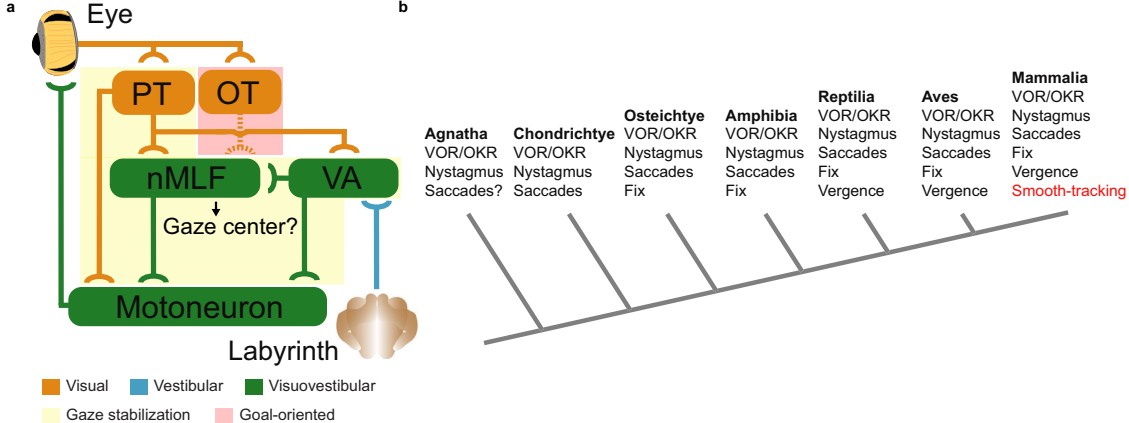

**Fig. 10 | Subcortical pathways controlling gaze-stabilization. a** A schematic showing the flow of visual (from the eye) and vestibular (from the labyrinth) information when producing gaze-stabilizing eye movements (shadowed in yellow) and the likely pathway underlying goal-oriented eye movements (shadowed in red). Brain areas that process only visual information are highlighted in orange, vestibular in blue, and visuovestibular in green. Note that, as in mammals, visual information already impacts vestibular inputs as soon as they enter the brain, and that all visuovestibular regions can be activated by visual or vestibular inputs independently. The motoneurons of the oculomotor nuclei initiate the VOR/OKR through recruiting the relevant extraocular muscles during the final step of the sensorimotor integration. **b** A phylogenetic tree featuring the seven main classes of vertebrates and the eye movements available to them[58,66]. Note that this diagram denotes the presence of an eye movement type within each class, meaning that not all member species are necessarily in possession of it. Smooth-tracking eye movements (in red) are present only in primates. The branches of the tree are not to scale. Abbreviations: PT pretectum, OT optic tectum nMLF nucleus of the medial longitudinal fasciculus, VA vestibular area.

a typical VOR seen across the vertebrate spectrum[56–58]. The velocity of saccadic movements in fish ranges between a hundred and a few hundreds of degrees per second[59–61]. Thus, saccadic eye movements are slightly slower in lampreys. In the lamprey, a functional cerebellum has not evolved, and the VOR can thus not have the complementary cerebellar circuit, which in other vertebrates is used for fine tuning of the VOR.

It is believed that saccades directed towards specific objects evolved from nystagmus. Lampreys possess a well-developed visual system and eye movements are evoked by non-optokinetic visual stimuli[24]. Thus, goal orienting saccades are likely present in these animals. Our results and the large similarities of the lamprey tectum with the mammalian superior colliculus[21–23,25] indicate that pretectum primarily controls gaze/posture-stabilization[54], whereas tectum drives goal-oriented eye movements (Fig. 10a). Together with a primordial visual cortex[26], it is likely that the main circuits controlling eye movements were already present before lampreys diverged from the main vertebrate line.

In conclusion, we have identified the oldest vertebrate example of OKR, which is integrated with VOR yielding enhanced eye-movement responses similar to mammals. Pretectum is the first level integrator of visual motion, subsequently projecting to vestibular and oculomotor nuclei like in mammals. We show that gaze-stabilization is fundamentally governed subcortically, allowing VOR-OKR interactions in the absence of cortex or cerebellum as well as locomotion-supported gaze-stability. In outlining nystagmus in lampreys, this study reveals the ancient origins of ballistic goal-oriented eye-movements, and shows that tectum likely downregulates gaze-stabilizing reflexes in favor of such commands. All eye movements are built from two basic types, slow and fast, both shown here to be present in lampreys. Thus, the neural template from which all eye movements arose was present already at the dawn of vertebrate evolution.

## Methods
### Animals
Experiments were performed on 50 adult river lampreys (*Lampetra fluviatilis*), and 6 young adult sea lampreys (*Petromyzon marinus*) of both sexes. The experimental procedures were approved by the local ethics committee (*Stockholms Norra Djurförsöksetiska Nämnd*) and the *Xunta de Galicia* under the supervision of the University of Vigo Committee for Animal use in Laboratory in accordance with the directive 2010/63/EU of the European Parliament and the RD 53/2013 Spanish regulation on the protection of animals use for scientific purposes. Animals were kept in aquaria with an enriched environment and continuously aerated and filtered water. Every effort was made to minimize suffering and to reduce the number of animals used.

### VOR eye movement recordings
To analyze the compensatory eye movements evoked by vestibular stimulation (VOR) in intact animals ($N = 9$), we used a transparent chamber with a video camera attached with a metallic arm that did not interfere in the visual field of the animal (Grasshopper3, GS3-U3-23S6M-C, FLIR Systems, Wilsonville). The diameter of the chamber was wide enough so that the animal could breathe normally but narrow enough to disallow swimming. The size of the chamber was based on average size data from several animal and those used for these experiments were chosen based on its size to fit under the previously described conditions in this chamber. The tube was filled with aerated cold water, and the animals mildly anesthetized with a dose of tricaine methane sulfonate (MS-222; 80 mg/L; Sigma-Aldrich) to facilitate their placement in the tube and minimize their stress. Once the animal recovered from the anesthetic, a series of quick vestibular stimulations in the roll, pitch and yaw planes were carried out in darkness while the evoked VOR eye movements were recorded. The total duration of the experiments was 2–3 min to minimize stress, and the animals were returned to their aquaria afterwards. In some of these experiments ($N = 3$) an accelerometer was attached to the chamber aligned with the vestibular organs, so that the kinematics of the vestibular stimulation could be retrieved and used to calculate the gain of the compensation performed by the lamprey eyes. To this effect, an ADXL335 accelerometer (Arduino, Sommerville, MA) was fitted to the rotating platform. This allowed us to retrieve the exact position of the lamprey at any given time, making it possible to compare head-and-eye positions.

To track eye movements, we used DeepLabCut 2.2[51], a Python software package which performs motion capture based on transfer learning with deep neural networks, and the obtained data were analyzed using custom Matlab R2020b scripts. To analyze the videos, first four labels were placed in the recorded eye in 20 frames chosen

randomly from each video so that the inferred trajectories could be averaged to minimize errors. Labels were also placed in the body to subtract the small movements originated from breathing and in response to the vestibular stimulation. The network was then trained, and once the training was evaluated, videos were analyzed to extract the trajectories of the labels and hence the eye. These data were used to extract the actual amplitudes of eye movements, and the diameter of the lamprey eye in the X and Y axes was measured in millimeters and then compared to the image resolution of the video recordings in pixels, providing a reliable conversion index. Eye-movement amplitudes recorded with the camera were consequently translated into millimeters. Having established the diameter of the lamprey eye, the angular displacement of VOR and OKR movements could then be retrieved based on trigonometric functions.

## Experimental preparation for visuo-vestibular stimulation

Placing animals in a transparent chamber generally saw them instinctively attaching their mouths to the chamber wall, stabilizing their head and allowing us to test VOR and nystagmus behaviorally. However, analyzing the relative contribution of visual and vestibular information required applying several experimental paradigms and could not be performed on intact animals. Thus, we decided to use an isolated preparation of the lamprey brain and rostral spinal cord with the eyes and vestibular organs. This allowed us to monitor eye movements via EMG recordings of the extraocular muscles to test visuovestibular integration, as well as record and inactivate different brain regions. For this, we first transected the head of animals deeply anesthetized with MS-222 (100 mg L$^{-1}$; Sigma), and then submerged it in ice-cooled artificial cerebrospinal fluid (aCSF) solution containing the following (in mM): 125 NaCl, 2.5 KCl, 2 CaCl$_2$, 1 MgCl$_2$, 10 glucose, and 25 NaHCO$_3$, saturated with 95% (vol/vol) O$_2$/5% CO$_2$. The dorsal skin and cartilage were removed to expose the brain, and the viscera and all muscles except the extraocular were removed to avoid movements, maintaining the eyes and otic capsules (where the vestibular organs are located) intact. To ensure that normal eye movements were preserved in the preparation and that the recorded EMGs in the extraocular muscles corresponded to actual eye movements, we performed electric stimulation in the AON at increasing intensities, monitoring eye movements and simultaneously recording in the dorsal rectus (Supplementary Fig. 1).

## Visuovestibular platform

To allow for extracellular and EMG recordings in response to coordinated visual and vestibular stimuli we built a tilting device that allowed vestibular stimulation of an eye-brain-labyrinth preparation (see above) in the roll plane, placed between two screens that allowed the coordinated presentation of visual stimuli. The platform was moved via a servo motor, and the angle and speed controlled with a microcontroller board (Arduino Uno). Visual stimuli were written in Matlab R2020b using the Psychophysics Toolbox Version 3 extensions[62,63] and another microcontroller board was subordinated to Matlab so that its outputs were used to coordinate the visual stimuli, tilting platform and electrophysiological recordings. The above-described preparation was pinned down in a transparent cooling chamber continuously perfused with aCSF at 6–8 °C, placed in the tilting platform inserted in a metallic cylinder connected with a Peltier plate to keep the temperature of the chamber. The preparation was aligned with the platform rotation axis so that vestibular stimulation was in the roll plane, avoiding translational movements, and facing the center of two screens placed at both sides at a preparation-screen distance of 9 cm, ensuring that the presented stimuli covered the entire visual field.

All experiments were carried out in darkness, so that the only source of light was the visual stimulation. Visual stimuli consisted of horizontal bars moving in the vertical axis with opposite directions on each screen (i.e. when bars presented to the right eye moved up to

down, bars in front of the left eye moved down to up). The direction of the bars was also adjusted to analyze yaw and pitch OKR responses.

To analyze visuovestibular integration, we applied only vestibular stimulation in the roll plane (with the screens turned off), only visual (preparation kept horizontal) and visuovestibular stimulation (tilting of the platform and combined visual stimulation). These paradigms were tested under four different conditions: low amplitude–low speed (5.8°; 48.7°/s), low amplitude–high speed (5.8°; 112.94°/s), high amplitude–low speed (22.7°; 48.7°/s), and high amplitude–high speed (22.7°; 112.94°/s). The angle refers to the maximum tilting of the platform, and the angular speed to the velocity of such tilting and/or the velocity of the optokinetic stimulation. Each of the paradigms (VIS, VES and VISVES) was applied three times on each animal under each condition (low amplitude–low speed, low amplitude–high speed, high amplitude–low speed, and high amplitude–high speed). The mean accelerations of the VES and VISVES protocols were 305.75 ± 112.79 deg/s$^2$ for the low velocity and 915.73 ± 324.58 deg/s$^2$ for the high velocity. This was calculated by attaching the accelerometer used in the behavioral studies to the platform and retrieving its motion dynamics. The recording electrode was placed in the dorsal rectus for roll and pitch recordings, and in the caudal rectus for yaw plane, and kept util termination of the experiment. After placing the electrode, the preparation was left to adapt for at least 30 min with a white screen which was used as a background for all the applied stimuli. 2 min were left between trials and presentation of VIS, VES and VISVES stimulations was randomized to minimize adaptation and compensate for possible changes in the excitability of the preparation. The number of repetitions performed on each animal was chosen to minimize the duration of the experiment while acquiring sufficient data, to ensure the viability of the preparation based on our experience in previous studies using similar isolated preparations[25–27]. EMG responses to the specific protocols generally exhibited similar dynamics for each animal. This allowed us to identify trials in which the neural integrity of the preparation was in any way damaged, in which case the experiment would be terminated.

## Extracellular recordings

To record muscle and/or neuronal activity, we used tungsten microelectrodes (-1–5 MΩ) connected to a differential AC amplifier, model 1700 (A-M systems). Signals were digitized at 20 kHz using pClamp (version 10.2) software. Tungsten microelectrodes were placed using a micromanipulator tightly fixed to the tilting platform, so that it rotated together with the preparation avoiding vibrations. In some cases, we also attached a video camera to the platform, to monitor the eye movements of the preparation.

To perform simultaneous recordings of the caudal and rostral extracellular eye muscles and a pair of ventral roots to assess gaze-stabilization during locomotion, we used a preparation exposing the brain together with the eyes and a large segment of the spinal cord. For this, we deeply anesthetized the animals with MS-222 (100 mgL$^{-1}$; Sigma) and transected the body 50–60 mm caudal to the second gill opening (roughly from the location of the obex). Then, submerged in ice-cooled artificial cerebrospinal fluid (aCSF), the dorsal skin and cartilage were removed to expose the brain and spinal cord, and the viscera and all muscles were removed. The optic nerves were sectioned, and the labyrinths removed to avoid visual and vestibular influences. Tungsten microelectrodes (-1–5 MΩ) were used to record muscle activity, while suction electrodes made from borosilicate glass (HilgenbergGmbH) using a vertical puller (Model PP-830; Narishige) filled with aCSF were used to record bilateral activity in the ventral roots of the spinal cord. Suction electrodes were connected to a differential AC amplifier, model 1700 (A-M systems).

Targeted lesions were carried out to identify the primary relay point for visual information between the eye and the extraocular muscle nuclei responsible for carrying out the OKR. To inactivate

tectum, the optic tract was sectioned just caudal to pretectum, eliminating the retinal input to this structure. To ensure that tectum was devoid of retinal fibers, Neurobiotin injections were performed at the end of the experiments in the optic tract at the level of pretectum, and brains were processed as described below for anatomical tract tracing. Labeling in the optic chiasm was used as a reference to confirm that the injection was performed in the optic tract. Pretectal lesions were performed by sectioning it acutely, and the same strategy was used to inactivate pallium. Pharmacological inactivation was also performed to inactivate pretectum while maintaining tectal integrity (see below).

### Anatomical tract tracing

Lampreys were deeply anesthetized with MS-222, and then transected at the level of the seventh gill. The head was submerged in aCSF solution and injections were made with glass micropipettes (borosilicate; o.d. = 1.5 mm, i.d. = 1.17 mm; Hilgenberg) with a tip diameter of 10–20 μm. Micropipettes were fixed to a holder attached to an air supply and a micromanipulator (model M-3333, Narishige), and 50-200 nL of Neurobiotin 20% (wt/vol) in aCSF containing Fast Green (Vector Laboratories) to aid visualization of the tracer was pressure injected in the oculomotor or the anterior/intermediate octavomotor nucleus. Following injections, the brains were kept submerged in aCSF in darkness at 4 °C for 24 h to allow transport of the tracers, and brains were then dissected out, fixed in 4% formaldehyde and 14% saturated picric acid in 0.1 M phosphate buffer (PB), pH 7.4, for 12–24 h, and cryoprotected in 20% (wt/vol) sucrose in PB for 3–12 h. Transverse sections (20 μm thick) were made using a cryostat and collected on gelatin-coated slides. For detection of Neurobiotin, Cy2 conjugated streptavidin (1:1000; Jackson ImmunoResearch) was used together with a deep red Nissl stain (1:500; Molecular Probes), diluted in 1% bovine serum albumin (BSA), 0.3% Triton X-100 in 0.1 M PB. Sections were mounted with glycerol containing 2.5% diazabicyclooctane (Sigma-Aldrich).

To label AON neurons projecting to the oculomotor nucleus for patch-clamp experiments, dextran amine-tetramethylrhodamine (3 kDa; 12% in saline; Molecular Probes) was pressure injected unilaterally into the AON tract projecting to nIII, at the level of the isthmic area. For this, the animals were deeply anesthetized with MS-222 (100 mg·L$^{-1}$) diluted in fresh water, and during the surgery and the injections the entire animal was submerged in aCSF containing MS-222 (80 mg·L$^{-1}$) to ensure that the animal was kept anesthetized. Then, an incision was done in the skin and the muscles directly above the rostral brainstem, and the cartilage was opened to expose the brain. Following injections, the dorsal skin was sutured, and the animal was returned to its aquarium for 48–72 h to allow transport of the tracer. Brains were then dissected out and processed for patch-clamp recordings (see below).

### Whole-cell recordings

Whole-cell current-clamp recordings were performed in thick slices maintaining pretectum for stimulation and exposing AON neurons for recording. For this, the entire brain was embedded in agar (4% in aCSF), and the agar block containing the brain was glued to a metal plate, quickly transferred to ice-cold aCSF and slices were cut using a vibrating microtome (Microm HM 650 V; Thermo Scientific). Afterward, the agar block was mounted in a submerged recording chamber.

Whole-cell current-clamp recordings were carried out using patch pipettes made from borosilicate glass (Hilgenberg GmbH) and obtained using a vertical puller (Model PP-830; Narishige). The resistance of the recording pipettes was 7–10 MΩ when filled with an intracellular solution with the following composition (in mM): 130 potassium gluconate, 5 KCl, 10 phosphocreatine disodium salt, 10 HEPES, 4 Mg-ATP, 0.3 Na-GTP; (osmolarity 265–275 mOsmol). In some cases, the electrode solution included 2 mM triethylammonium

bromide (QX314; Sigma-Aldrich) to block action potentials. Bridge balance and pipette-capacitance compensation were adjusted for using a MultiClamp 700B patch amplifier and Digidata 1322 analog-to-digital converter under software control 'PClamp 10.2' (Molecular Devices). The preparation was constantly perfused with aCSF at 6–8°.

Stimulation of the pretectum/optic tract was performed with the same borosilicate glass microcapillaries used for patch recordings, connected to a stimulus isolation unit (MI401; Zoological Institute, University of Cologne). The stimulation intensity was set to one to two times the threshold strength (typically 10–100 μA) to evoke PSPs.

### Eye and body tracking in a semi-intact preparation

To analyze eye movements in response to swimming without visual and vestibular influences, we used a semi-intact preparation exposing the brain and eyes while leaving the rest of the body intact. For this, the animals were anesthetized with MS-222 (100 mg·L$^{-1}$) diluted in fresh water, and the dorsal skin, muscles and cartilage were removed in the head of the animal, so that the brain, otic capsules, and eyes were exposed to allow the dissection of the optic nerves and labyrinths, and the monitoring of the eyes. The preparation was left to recover, and spontaneous and evoked swimming episodes (by gently pinching the tail with forceps) were recorded, together with eye movements, using a video camera coupled to the microscope. To ensure that the oculomotor system was intact, eye movements were monitored for all preparations both spontaneous and evoked by electric stimulation of the optic tract. Eyes and tail movements were tracked using DeepLabCut as described above. Four labels were placed on each eye, and two labels were placed in the rostral body of the animal to average data and therefore minimize errors in the inferred trajectories of the eye and the body.

### Yaw platform for nystagmus analysis

To apply large amplitude vestibular stimulations in the yaw plane, we developed a platform moved by a servo motor controlled via Arduino, so that the speed and amplitude could be controlled. As for VOR recordings (see above), a transparent chamber with the appropriate size to avoid that the animal could swim was fixed on the rotating platform, filled with aerated cold water. The head of the animal was aligned with the axis of rotation to avoid translational movements, and a video camera (Grasshopper3, GS3-U3-23S6M-C, FLIR Systems) was placed facing one of the eyes. 180° or 360° rotations were applied at different speeds, and the eye movements were tracked using DeepLabCut and analyzed using custom Matlab scripts.

### Drug applications

To test the role of pretectum in mediating OKR keeping tectal integrity during EMG recordings, the glutamate receptor antagonist kynurenic acid (4 mM; Sigma-Aldrich) was locally applied in the pretectum by pressure injection through a micropipette fixed to a holder (containing Fast Green to aid visualization of the injection spread), which was attached to an Picospritzer-II Microinjection Dispense System (Parker). The holder was connected to a micromanipulator to monitor the position of the pipette and ensure precise drug injections. To evoke fictive locomotion while monitoring eye movements vis EMG recordings, we used a split chamber separating the exposed spinal cord from the brain, and the N-methyl-D-aspartate receptor agonist D-Glutamate (0.75 mM, Sigma-Aldrich) was bath applied to the spinal cord.

### Image analysis

Photomicrographs were taken using a digital camera (Olympus XM10) mounted on an Olympus BX51 fluorescence microscope. Illustrations were made using Adobe Illustrator CC 2019 and GIMP 2.1 (GNU image manipulator program). Images were only adjusted for brightness and contrast. Confocal Z-stacks of optical sections were obtained using a Zeiss Laser scanning microscope 510, and the projection images were processed using the Zeiss LSM software, ImageJ 1.53k and GIMP 2.1.

## Quantification

For all electrophysiological recordings, data analysis was performed using custom written functions in Matlab. For video recordings, positions were extracted using DeepLabCut[30]. To calculate gain during VOR stimulation, two different strategies were used. On one hand, position gain was calculated as the ratio of the areas under the curve of head and eye positions for a specific time interval. For dynamic gain, the ratios between eye and head velocities were calculated between two different points[64]. In both cases, gain was calculated during the slow phase of VOR, avoiding nystagmic quick phase movements. To quantify the duration of the nystagmus quick and slow phases, analysis was done frame by frame using Adobe Premiere. This allowed us to identify the exact starting and ending position of any eye movement with a high level of accuracy, with the time points being recorded to calculate the duration and the amplitudes measured in pixels and then converted in degrees as exposed above. Amplitudes were calculated measuring the distance in pixels traveled by the eye between these key frames, and then converted to angles as exposed above.

For EMGs and extracellular recordings, the number of spikes was quantified to be compared after different conditions. Maximum amplitudes were also measured: The signals were fully rectified and the distance from baseline to peaks was calculated as a way to measure the amplitudes. Given that isolated units cannot be extracted in EMGs obtained in response to electric stimulation of the eye, the integral under the curves was compared after fully rectifying the signals using trapezoidal numerical integration ('trapz' function), as a way of integrating the amplitude and duration of the signals. For whole recording analysis, PSPs amplitudes were measured after the synaptic decay was fitted by an exponential curve to extract correct amplitudes, since subsequent PSPs often started on the decay phase of previous responses. The responses recorded in each animal were highly consistent for between all repetitions applied of each paradigm (VIS, VES or VISVES). However, differences were observed among animals due to the location of the electrode and the excitability of the preparation. Therefore, data were normalized, dividing all values to the maximum response in each animal and for each modality. This way, the ratio for each sensory modality was obtained for each animal and each experimental condition (low amplitude–low speed, low amplitude–high speed, high amplitude–low speed, and high amplitude–high speed). Data were then averaged among animals, obtaining the mean contribution of each sensory modality represented in the plots.

## Statistics and reproducibility

For statistical analysis, all traces were pooled for each dependent and independent variable. Analysis was done using SPSS 25 and JASP 0.16. In some cases, spontaneous eye movements were generated immediately before or after the stimulation, clearly affecting the responses analyzed. After visually inspecting putative outliers, a Grubb's test was used to identify significant outliers, and data points outside of the 95% confidence interval were removed prior to performing the statistical analysis. Paired T-tests were used to compare signal amplitudes and number of spikes between modalities at $\alpha = 0.05$. Throughout the figures, sample statistics are expressed as means ± SD. Values falling outside the 95% confidence interval according to Grubb's test were excluded for plots. For analyzing visual ramps of roll and pitch planes, a repeated measures ANOVA was implemented with the respective stimulation increments as factors. Linear regression analyses were used for the behavioral trials, retrieving Pearson's coherence coefficient to test the conjugacy between the two eyes as well as between the average eye movements and that of the tail; for the latter, eye positions were offset by seven frames forward in time so as to match the onset of the tail movement. The number of experiments perfomed for which representative examples are shown is as follows: For gain analysis (Fig. 1b–d), vestibular stimulation in the three planes was applied to

three animals. Three repetitions for each plane stimulation were analyzed per animal. Neurobiotin injections were performed in the nIII of three animals (Fig. 5), and in the AON of four animals (Fig. 7b–f). Nystagmus experiments were performed in three animals (Fig. 8). The experiments to analyze the presence of locomotion-evoked eye movements in a semi-intact preparation were performed in six animals (Fig.9). Statistical significance is shown as follows: *$P < 0.05$, **$P < 0.01$, ***$P < 0.001$.

### Reporting summary

Further information on research design is available in the Nature Research Reporting Summary linked to this article.

## Data availability

Source data are provided with this paper and can be downloaded in the following link (https://doi.org/10.5281/zenodo.6628365), together with additional raw data[65]. Further information and requests should be addressed to the corresponding author.

## Code availability

The code used for analysis and presenting the visual and vestibular stimuli in coordination with electrophyisiological recordings can be downloaded in the following link: (https://doi.org/10.5281/zenodo.6628365)[65]. Further information and requests should be addressed to the corresponding author.

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

## Acknowledgements

We thank Professor Abdel El Manira and Dr. Brita Robertson for valuable comments on the manuscript, Roberto de la Torre for the video camera, and Roi Carrera Boo for setting up DeepLabCut. This work was supported by the Swedish Medical Research Council (VR-M-K2013-62X-03026, VR-M-2015-02816, VR-M-2018-02453 to S.G., and VR-M-2019-01854 to J.P.-F.), Proyectos I + D + i PID2020-113646GA-I00 funded by MCIN/AEI/ 10.13039/501100011033 and by "ERDF A way of making Europe" (to J.P.-F.), the Ramón y Cajal grant RYC2018-024053-I funded by MCIN/AEI/ 10.13039/501100011033 and by "ESF Investing in your Future" (to J.P.-F.), Xunta de Galicia (ED431B 2021/04 to J.P.-F.), EU/FP7 Moving Beyond grant ITN-No-316639, European Union Seventh Framework Programme (FP7/2007-2013) under grant agreement no.604102 (HBP), EU/Horizon 2020 no.720270 (HBP SGA1), no. 785907 (HBP SGA2) and no. 945539 (HBP SGA3) to SG, the CINBIO, the Gösta Fraenckel Foundation for Medical Research FS-2020:0004 (to T.W.), the Sigvard and Marianne Bernadotte Research Foundation for Children's Eye Care, and the Karolinska Institutet.

## Author contributions

T.W. and J.P.-F. conceived the study and designed and carried out the experiments. S.G. contributed towards the theoretical framework, provided critical feedback and supplied equipment and materials. T.P. provided financial support and supervision. J.P.-F. designed the figures with input from T.W. T.W. and J.P.-F. performed the analysis, interpreted the results and wrote the manuscript with input from all authors.

## Competing interests

The authors declare no competing interests.
