## [Peer Review File · Nature Communications]

Conserved subcortical processing in visuo-vestibular gaze controlREVIEWER COMMENTS

Reviewer #1 (Remarks to the Author):

The work of Wibble et al described what they claim 'all basic components of the vesio-vestibular control of gaze were presented in all vertebrates'. The work is well documented nearly all the relevant citations and is overall an important publication. I suggest expanding the lamprey story by citing the work showing the absence of the horizontal canal and has a compensation by the anterior and posterior crista input, an interesting story on its own right. Likewise, the formation of three III MNs have to split to become four III MNs that is in contrast to the roll position provided by the contralateral MN in all vertebrates.

Suggested changes:

p39 suggest adding the work of Straka et al, 2014

p46 suggest citing the work of Chagnaud et al., 2017

p49; No, the similarity is different: it is driven by two VI innervation in lampreys that leaves only 3 III innervations left. I suggested to fully describe the proper description of lampreys (see Fritzsich, B., 1998. Evolution of the vestibulo-ocular system. *Otolaryngology—Head and Neck Surgery*, 119(3), pp.182-192. And Chagnaud, B.P., Engelmann, J., Fritzsich, B., Glover, J.C. and Straka, H., 2017. Sensing external and self-motion with hair cells: a comparison of the lateral line and vestibular systems from a developmental and evolutionary perspective. *Brain, behavior and evolution*, 90(2), pp.98-116.

p57 Yes, in general correct but lack details that show a different retina innervation (see Fritzsich and Collin, 1990)

p57 The vestibular system is different from lampreys: The receive only input from the anterior and posterior cristae (Fritzsich, B., Signore, M. and Simeone, A., 2001. *Otx1* null mutant mice show partial segregation of sensory epithelia comparable to lamprey ears. *Development genes and evolution*, 211(8-9), pp.388-396.) that receives a third 'dorsal cristae' that is unclear on its central projection (Fritzsich, 1998; Chagnaud et al., 2017). The details of the vestibular input required an expansion of the vestibular connections that lacks the yaw input and the horizontal crista.

p107 Yes, the dorsal rectus is the only similarity to all lamprey eye muscles (see Fritzsich et al., 1990 that migrates to the contralateral motoneurons (Cheng, L., Desai, J., Miranda, C.J., Duncan, J.S., Qiu, W., Nugent, A.A., Kolpak, A.L., Wu, C.C., Drokhyansky, E., Delisle, M.M. and Chan, W.M., 2014. Human CFEOM1 mutations attenuate KIF21A autoinhibition and cause oculomotor axon stalling. *Neuron*, 82(2), pp.334-349.; and Jahan, I., Kersigo, J., Elliott, K.L. and Fritzsich, B., 2021. Smoothened overexpression causes trochlear motoneurons to reroute and innervate ipsilateral eyes. *Cell and Tissue Research*, 384(1), pp.59-72.;

p112 Yaw would be particularly interesting, given that the horizontal canal does not exist in lampreys (see Fritzscht et al., 2001).

p192 cite the work of Fritzscht, 1998 that describes the central vestibular input to the IIIrd MNs.

p205 see the publication of Fritzscht 1998 that shows the contralateral anterior octavo-motors.

p335 We have demonstrated that lampreys have no horizontal canal and needs to compensate the vestibular input from a newly formed gnathostomes. Given that the yaw similarity suggest a parallel evolution to the newly formed input. Please be clear about similarity in function that is not identical to the lamprey vestibular and oculomotor system.

p446 full citation is need for Rovainen 1976

Fig 1 The yaw presentation is a different movement that shows an interruption of sorts. Given that the horizontal canal is not form in lampreys (see Fritzscht,1998, Fritzscht et al., 2001) the compensation should happen with the anterior and posterior cristae. How should that compensate happens is unclear.

Fig 2. Indeed there is no yaw response (G). Consistent with our previous tracing set of data.

Fig 3 Tracing was shown by Fritzscht, 1998 demonstrating the AON connection to the III MNs

Fig. 8 Once we add the horizontal canal and add a fourth oculomotor in gnathostomes we end up with a different yaw input in lamprey that compensates for the absence of the horizontal input. With these addition we add that likely similarity to yaw input.

Reviewer #2 (Remarks to the Author):

This paper investigates both the vestibulo-ocular reflex (VOR) and optokinetic reflex (OKR) in the lamprey, a vertebrate with ancient origins in evolution. The authors provide evidence that both reflexes are present, and that the reflexes are served by circuitry with notable similarities to vertebrates of more recent origin.

I commend the authors on showing that two vital components of the vertebrate ocular motor system have ancient roots. Unfortunately, the paper is marred by deficiencies in experimental design, sample size, data analysis, presentation of results, and clarity of description. These problems are as follows.

Experimental design. The authors frequently refer to gaze stabilisation. For example, line 325 of the Discussion states that “our findings push the onset of visually guided gaze-stabilisation further back by 150 million years”, and line 330 adds that the vestibulo-ocular reflex in lamprey “reliably compensates for head movements in all three planes”. The paper, however, provides no direct evidence for gaze stabilisation. To do that the authors would have to quantify eye movements and show the extent to

which they compensate for head and stimulus movement. Instead, much of the evidence consists of intramuscular EMG from one eye muscle during vestibular and visual stimulation. These EMG records are a very limited indicator of eye movement. The authors need to be more careful in their claims. Given that the equipment includes an eye tracker, it would greatly assist the reader to know why it was not used to measure the extent to which gaze is stabilised.

Figures 2, 3, 4. There are problems with the presentation of EMG responses. First, the EMG time courses are bursty and variable in amplitude. Normalisation of responses is therefore dangerous because it combines strong with weak responses without distinguishing between them. It would be better to show a mean unnormalised response along with either individual samples or confidence intervals. Second, the method of normalisation is unclear. Where are the voltage calibrations for individual responses? Why do points never reach a normalised value of 1 in most graphs? How is “amplitude” measured? The number of samples is also often unclear. For example, where multiple recordings were made in the same animal, what was the total time interval, and were both sides of the animal used?

Statistics. Statistical tests in the paper are frequently reported as a p-value with little further description. This is insufficient to judge the quality of the test and result. The following should be provided for every statistical test: name of test, test statistic, number of samples or degrees of freedom, the p-value.

Role of pretectum in OKR. The authors maintain (line 23 of the Abstract) that the OKR is mediated by the pretectum. This assertion appears to rely on a single recording in a single animal – Figure 4h. This demonstration is inadequate. To support their claim, they need to show that the OKR is present before and after, but not during, pretectal inactivation in a statistically significant number of recordings across several animals.

Animal preparation. The study largely depends on an isolated eye-brain-labyrinth preparation. It was never clear to me why this was preferable to using a minimally invasive approach to the whole animal. Preparation of isolated tissues runs the risk of indirect damage to those tissues, and deterioration over time. The paper needs to justify the use of an isolated preparation, and to show that critical properties such as eye movements were normal. The authors also need to indicate how they measured tissue deterioration over time, and criteria used to terminate an experiment.

Methods. The Methods section is often unclear. For example, lines 815 and 854 state that “all muscles were removed”. How, then, were EMGs recorded? I suggest that at least two of the authors review this section to improve its clarity.

There were also some less serious problems with the paper, as follows.

- Line 968. Outliers were removed. This is dangerous because examination of the outliers could reveal previously unappreciated factors in the experiment. Outliers should be described, and their removal should be justified.
- Vestibular stimulation is transient. Given that a change from one rotation speed to another cannot be instantaneous, the time course of acceleration should be described.
- A positive note! The nystagmus in Figure 6 is fascinating, even more so because it was shown in the whole animal. How does the speed of the quick phase compare with that in other cold-water animals, and how well does the eye movement compensate for head rotation?

Reviewer #3 (Remarks to the Author):

This manuscript examines the visual mechanisms of gaze-stabilization in the lamprey, a member of the most ancient group of vertebrates. Using an ingenious platform to conduct electrophysiological recordings while measuring eye movements and providing vestibular and visual stimulation, this study investigates the influence of visual signals on the vestibulo-ocular reflex and the underlying neural pathways. Behavioral data show visuomotor responses and a response pattern reminiscent of optokinetic nystagmus. Electrophysiological results show strong visuo-vestibular integration and identify various areas involved in this process. I find these experiments well designed and the results interesting. There are, however, various points in the text in which things are not well explained and need to be refined.

General comments

Nystagmus. One aspect that is not provided in the manuscript is a characterization of the type of behavior elicited by visual stimulation. Part of the problem in understanding this is also that the text does not clearly explain in which way the stimulus and the platform move. What is causing the cyclical behavior of the traces in Figure 1, a sinusoidal oscillation of the platform or a sort of optokinetic nystagmus? One would assume a OKN-type of behavior to be visible in the absence of VOR, and perhaps this is the function of Figure 1a. But things remain unclear both because the experimental conditions are not well explained and because examination of nystagmus is postponed to much later (Figure 6), using a different preparation. In addition to clarifying these issues, it would be helpful to see if the resetting movements---with or without VOR---do resemble saccades, by reporting their main sequence characteristics (the relationships between peak speed, duration, and amplitude).

Corollary discharges and yaw movements. The section on lines 292-314 is very confusing. To start, the driving hypothesis is not clear: why the presence of non-visually driven compensatory eye movements

would justify the absence of OKR on the yaw plane? The conclusions seem contradictory: it is first stated that compensatory movements were only observed in one out of four animals (line 302; why proceeding if this behavior is not typical?). And later concluded that no compensatory responses were evoked in the eye muscles (line 312; not even in the one animal that did exhibit a compensatory behavior? How could this be?). I must be missing something here, because the entire section does not seem to make much sense.

Figures. Another general comment regards the description of the figures, which are overall nice, but often contain elements that are not explained neither in the text nor caption. Several specific examples are provided below.

Specific comments

Figure 1. (1A) What are the two columns of eye position? Please insert labels on top. And the circles and dashed lines in each panel? (1F) I find the way of marking visual stimulation via stripes difficult to read, as it overlaps with the data and other event markers. Why not adding temporal sequence of events as bars on top/bottom?

Figure 2. Why are EMG amplitudes and spikes plotted in normalized coordinates? The text does not explain what information is gained from either of these two metrics individually. The amplitude is relegated to a supplementary figure in Figure 3, but not clear why. Simply because it didn't yield a significant result?

Figure 3. The characteristics of motion are not clear. I assume that here amplitude and speed refer to the peak angle of the platform and speed of motion of the visual stimulus, but the text does not explain. Part of the problem is that I would expect the platform to continually rotate to study VOR, but panel F seems to state that the platform only moves briefly at the very beginning (the blue stripe; note that legends are not defined here). Is the platform moving or not?

A further thing to pay attention to is the use of the term amplitude to describe the movement characteristics, creating possible confusion with the amplitude of the EMG signals.

Line 162. Not clear why the authors conclude that visual stimulation plays a more important role at the beginning of compensatory eye movements.

Line 223: Supplementary Figure 3i not 3b

Line 268: Maybe Figure 5i, rather than 5g?

REVIEWER COMMENTS

Reviewer #1 (Remarks to the Author):

The work of Wibble et al described what they claim 'all basic components of the vesio-vestibular control of gaze were presented in all vertebrates'. The work is well documented nearly all the relevant citations and is overall an important publication. I suggest expanding the lamprey story by citing the work showing the absence of the horizontal canal and has a compensation by the anterior and posterior crista input, an interesting story on its own right. Likewise, the formation of three III MNs have to split to become four III MNs that is in contrast to the roll position provided by the contralateral MN in all vertebrates.

We thank the reviewer for the positive comments and suggestions. We agree that expanding what it is known about the lamprey vestibular and oculomotor systems adds strength to the paper. We were however very constrained in our initial submission by space limitations and number of references (only fifty allowed), which made it impossible to include several relevant aspects in the manuscript. The reason for this was that the manuscript was initially submitted to *Nature Neuroscience* and then we were invited to transfer it to *Nature Communications* without reformatting. However, we have now less restrictions (500 more words and 70 references) and have included additional references as suggested by the reviewer and discussed the different aspects raised by the reviewer, as we respond below point-by-point.

Suggested changes:

p39 suggest adding the work of Straka et al, 2014

We agree with the reviewer that this work offers valuable context for the sentence, and we have added this.

p46 suggest citing the work of Chagnaud et al., 2017

We agree that this reference is relevant, discussing developmental and evolutionary aspects of the lateral line and vestibular system, and have included it in the new paragraph related to the vestibular system (see below).

p49; No, the similarity is different: it is driven by two VI innervation in lampreys that leaves only 3 III innervations left. I suggested to fully describe the proper description of lampreys (see

Fritzsch, B., 1998. Evolution of the vestibulo-ocular system. *Otolaryngology—Head and Neck Surgery*, 119(3), pp.182-192. And Chagnaud, B.P., Engelmann, J., Fritzsch, B., Glover, J.C. and Straka, H., 2017. Sensing external and self-motion with hair cells: a comparison of the lateral line and vestibular systems from a developmental and evolutionary perspective. *Brain, behavior and evolution*, 90(2), pp.98-116.

We agree with the reviewer that it is relevant to include a more detailed description of the lamprey oculomotor and vestibular systems. We have now included a paragraph adding more details about the vestibular system expanding the line 57, and another one discussing the oculomotor system (see below), in which we include the suggested references. Regarding this sentence (line 49), to reflect similarities but also differences, we have now rephrased it to (line 48):

“The VOR, on the other hand, appeared very early during vertebrate evolution and, although different species-specific configurations may exist, the disynaptic circuit that converts vestibular information in the appropriate compensatory eye movement is present in all vertebrates.”

p57 Yes, in general correct but lack details that show a different retina innervation (see Fritzsch and Collin, 1990)

The sentence of line 57 refers primarily to lampreys possessing well-developed eyes and extraocular muscles. We use these studies as background for our hypothesis that the animals may be capable of producing an OKR. The suggested article, outlining the dendritic arborization of ganglion cells and retinopetal fibers in the retina of the silver lamprey, is of great importance for our understanding of the visual system and shows that retinal organization shows some differences when compared to that of gnathostomes, suggested to be a primitive organization. Although this differences in retinal organization may have an impact on motion processing, we feel that adding these details is out of the scope of the paper. We have however rephrased the sentence and included the suggested article as follows (line 57):

“Although with small differences in retinal organization when compared to gnathostomes (Fritzsch and Collin, 1990), lampreys have well developed image-forming camera eyes, and the organization of eye muscles and motor nuclei is remarkably similar to other vertebrates.”

p57 The vestibular system is different from lampreys: The receive only input from the anterior and posterior cristae (Fritzsch, B., Signore, M. and Simeone, A., 2001. *Otx1* null mutant mice show partial segregation of sensory epithelia comparable to lamprey ears. *Development genes*

and evolution, 211(8-9), pp.388-396.) that receives a third 'dorsal cristae' that is unclear on its central projection (Fritzschn, 1998; Chagnaud et al., 2017). The details of the vestibular input required an expansion of the vestibular connections that lacks the yaw input and the horizontal crista.

We agree that further discussing these aspects is relevant for the article. It is true that the vestibular system is different in lampreys in that they only have two semicircular canals, whereas vestibular information in the yaw plane is coded by two horizontal ducts thought to have evolved in parallel (Makland et al., 2014). However, the anterior and posterior canals show clear homologies with their gnathostome counterparts (Makland et al., 2014; Higuchi et al., 2019). In this sentence we specifically refer to the disynaptic arc that underlies VOR that is present in all vertebrates, independently of species-specific arrangements of how information in one plane is conveyed to the corresponding set of extraocular muscles.

We agree that this perspective, and the background of the sensory systems, may benefit from added clarity and context. To illustrate what is known about the lamprey vestibular and oculomotor systems and what is known about the evolution of the VOR, we have now included two paragraphs, one in the introduction about the vestibular system, and another in the discussion about both the vestibular and the oculomotor systems:

Paragraph in the Introduction expanding these aspects (line 60):

"They exhibit VOR and a well-developed vestibular system, featuring a labyrinth with two semicircular canals (anterior and posterior) considered homologous to their gnathostome counterparts, while lacking a lateral horizontal canal (Fritzschn, 1998; Makland et al., 2014; Higuchi et al., 2019). However, lampreys have a horizontal duct system that provides vestibular information also in the yaw plane, which seems to have evolved in parallel to the gnathostome horizontal canal (Fritzschn et al., 2001; Makland et al., 2014; Higuchi et al., 2019). The lamprey labyrinth is therefore capable of generating compensatory eye and body movements in the yaw plane using the same brain circuit architecture responsible for roll and pitch rotations."

In the Discussion we added the following paragraph (line 388):

"The available data suggests that the disynaptic arch underlying the VOR appeared in early vertebrates supported by the anterior and posterior semicircular canals and that it has been largely maintained in all vertebrate groups (Straka and Baker, 2013). The appearance of a lateral horizontal canal in gnathostomes, which is Otx dependant (Fritzschn, 1998; Fritzschn et al., 2001), gave rise to some circuit arrangements (Fritzschn, 1998). Namely, the six extraocular

muscles exhibited by the lamprey are thought to be homologous of those of bony fishes and tetrapods (Fritzscht et al. 1990; Fritzscht, 1998), but with some differences in their motor nuclei innervation as compared to gnathostomes. Three muscles are innervated by the oculomotor nucleus (nIII), one is innervated by the trochlear nucleus (nIV), and two by the abducens (nVI). In elasmobranchs four muscles are innervated by the nIII, while one is innervated by the nIV and one by nVI. In bony fish and tetrapods four muscles are innervated by the nIII, one by the nIV, and at least one by nVI (Fritzscht, 1998; Jahan et al., 2001). However, the same computational scheme, also present in lampreys, was maintained. Interestingly, lampreys exhibit an overall pattern of second-order vestibular projections very similar to mammals (Fritzscht, 1998). The vestibular system thus shows highly conserved elements as well as variations related to specific functional variations (Straka and Baker, 2013; Chagnaud et al., 2017).”

p107 Yes, the dorsal rectus is the only similarity to all lamprey eye muscles (see Fritzscht et al., 1990 that migrates to the contralateral motoneurons (Cheng, L., Desai, J., Miranda, C.J., Duncan, J.S., Qiu, W., Nugent, A.A., Kolpak, A.L., Wu, C.C., Drokhlyansky, E., Delisle, M.M. and Chan, W.M., 2014. Human CFEOM1 mutations attenuate KIF21A autoinhibition and cause oculomotor axon stalling. *Neuron*, 82(2), pp.334-349.; and Jahan, I., Kersigo, J., Elliott, K.L. and Fritzscht, B., 2021. Smoothed overexpression causes trochlear motoneurons to reroute and innervate ipsilateral eyes. *Cell and Tissue Research*, 384(1), pp.59-72.;

As exposed in the previous comment, we have now expanded the information about similarities and differences, including most of the references suggested by the reviewer.

p112 Yaw would be particularly interesting, given that the horizontal canal does not exist in lampreys (see Fritzscht et al., 2001).

Given the reliable responses obtained in the roll and pitch planes we were very surprised to not get optokinetic responses in the yaw plane. As the reviewer mentions there are anatomical differences in the lamprey ear related to the yaw plane (the absence of the lateral semicircular canal that appears in gnathostomes as we have now referenced in previous comments). Still, eye movements are reliably evoked in response to vestibular stimulation in the yaw plane, and the differences in the vestibular apparatus should not be related to the visual processing and therefore to the absence of optokinetic responses in the yaw plane. There is clearly a need for studies focusing in on this lack of a yaw OKR, and we look forward to investigating it further in the future.

p192 cite the work of Fritzs, 1998 that describes the central vestibular input to the IIIrd MNs.
We agree that the suggested study is relevant, and it is now included this reference.

p205 see the publication of Fritzs 1998 that shows the contralateral anterior oculo-motors.
We have included the reference.

p335 We have demonstrated that lampreys have no horizontal canal and needs to compensate the vestibular input from a newly formed gnathostomes. Given that the yaw similarity suggest a parallel evolution to the newly formed input. Please be clear about similarity in function that is not identical to the lamprey vestibular and oculomotor system.

We agree that there may be some ambiguity in the sentence relating to this matter. "Here we confirm that disynaptic pathways underlying VOR in lampreys are similar to other vertebrates" refers to the basic circuit that transforms vestibular information into compensatory eye movements that is common to all vertebrates (from the vestibular organ to vestibular nuclei in the rhombencephalon which in turn project to oculomotor nuclei), rather than to projections related to specific planes and/or extraocular muscles that are variable among species.

To clarify this, we have rephrased this sentence to (line 383):

"Here we confirm that the basic circuit underlying VOR via disynaptic pathways was already present in lampreys and thereafter maintained through vertebrate evolution albeit with species-specific modifications"

p446 full citation is need for Rovainen 1976.

This oversight has been corrected.

Fig 1 The yaw presentation is a different movement that shows an interruption of sorts. Given that the horizontal canal is not form in lampreys (see Fritzs,1998, Fritzs et al., 2001) the compensation should happen with the anterior and posterior cristae. How should that compensate happens is unclear.

The movements observed in the figure are not interruptions, but active movements in the opposite direction of the slow phase of VOR. They correspond to nystagmus beats that we further analyze in Figure 6, showing that their features corroborate their nystagmus identity (see comments below). We also observed them in the roll plane, and to less degree in the pitch plane. However, it is extremely difficult to apply full rotations in these planes to intact animals, and thus we decided to investigate nystagmus dynamics in the yaw plane.

Fig 2. Indeed there is no yaw response (G). Consistent with our previous tracing set of data. In figure 2G we show lack of responses in the yaw plane after applying visual stimulation only (optokinetic) with no vestibular contribution. We understand that the reviewer suggests that the differences in the lamprey labyrinth with respect to other vertebrates could predict somehow the lack of optokinetic responses in the yaw plane, because we are not aware of any previous study in lampreys investigating OKR and showing that there are no optokinetic responses in the yaw plane or providing anatomical (tracing) data that would suggest this to be the case.

Fig 3 Tracing was shown by Fritzsich, 1998 demonstrating the AON connection to the III MNs We have now included this reference in the text, as outlined in the previous comment relating to this study. Regarding figure 3, we show this projection in adult lampreys, whereas in Fritzsich et al., (1998) and in the cited Pombal et al., (1996) the AON connection to nIII was shown in larvae, and this is what we indicated in the text (lines 226-227):

“Some of the pathways involved in VOR have been analyzed in lamprey larvae, but no data in adults is available”.

Fig. 8 Once we add the horizontal canal and add a fourth oculomotor in gnathostomes we end up with a different yaw input in lamprey that compensates for the absence of the horizontal input. With these addition we add that likely similarity to yaw input.

As mentioned in a previous comment relating to the horizontal canal, we have included additional context for these issues in the manuscript. In figure 8 we summarize the overall connections underlying VOR/OKR, illustrating how visual and vestibular inputs are conveyed to different brain nuclei underlying gaze-stabilization. This is a general scheme showing the flow of sensory information without focusing on how vestibular information on a single plane is conveyed to the corresponding extraocular muscles. An in-depth construct involving the specific canals and their pathways may indeed be of great interest when discussing the evolutionary trajectories of these structures. We do however feel that this manuscript may not be the place for such a schematic, as we aim to elucidate the central brain regions common for the sensory integration of vestibular and visual inputs.

Reviewer #2 (Remarks to the Author):

This paper investigates both the vestibulo-ocular reflex (VOR) and optokinetic reflex (OKR) in

the lamprey, a vertebrate with ancient origins in evolution. The authors provide evidence that both reflexes are present, and that the reflexes are served by circuitry with notable similarities to vertebrates of more recent origin.

I commend the authors on showing that two vital components of the vertebrate ocular motor system have ancient roots. Unfortunately, the paper is marred by deficiencies in experimental design, sample size, data analysis, presentation of results, and clarity of description. These problems are as follows.

We thank the reviewer for stating the relevance of the study and for the suggestions. We have now addressed all the issues and included new experiments that we detail point by point.

Experimental design. The authors frequently refer to gaze stabilisation. For example, line 325 of the Discussion states that “our findings push the onset of visually guided gaze-stabilisation further back by 150 million years”, and line 330 adds that the vestibulo-ocular reflex in lamprey “reliably compensates for head movements in all three planes”. The paper, however, provides no direct evidence for gaze stabilisation. To do that the authors would have to quantify eye movements and show the extent to which they compensate for head and stimulus movement. Instead, much of the evidence consists of intramuscular EMG from one eye muscle during vestibular and visual stimulation. These EMG records are a very limited indicator of eye movement. The authors need to be more careful in their claims. Given that the equipment includes an eye tracker, it would greatly assist the reader to know why it was not used to measure the extent to which gaze is stabilised.

As the reviewer points out, the ideal experimental approach would be to quantify eye movements using tracking in intact animals. However, as we discuss in the comment below, it is impossible to maintain the animals in the right conditions long enough to apply the three visuovestibular paradigms (VIS, VES, and VISVES). However, realizing the importance of behavioral demonstrations, we did experiments in which it was possible to quickly get results with intact animals (Figures 1 and 6) allowing us to prove behaviorally that eye movements are evoked when the animals are rotated in the three planes and that they have nystagmus.

The reviewer is right in that it is important to provide evidence for gaze stabilization. We have therefore performed a new set of experiments in intact animals quantifying the extent of eye movement compensation in response to vestibular stimulation in the three planes. For this, we measured head movements by using an accelerometer and quantified the position gain as the ratio between the areas under the curve for eye and head position at a given moment, as well

as the dynamic gain as the ratio of slow phase velocities. Our results show that the VOR dynamic gain is around 0.7, and the position gain is around 0.6, demonstrating that lampreys exhibit VOR gaze stabilization. These values were obtained after quick stimulations (~60-200°/s) and were aimed to proving that lampreys stabilize gaze. For proper gain analysis, we should carry out a detailed analysis analyzing VOR gain in response to a full range of velocities, which we think should be carried out in a separate study.

These results were incorporated in the text as follows (line 94):

“In order to investigate the gain of the eye in relation to the head movement an accelerometer was fitted to the transparent tube (N=3). The tube was then maneuvered in the three planes manually at velocities ranging between 60-200 deg/s while the lamprey eye was tracked. Eye and head positions were subsequently plotted (Fig. 1b-d) and compared in terms of their spatial and temporal alignment (Supplementary Fig. 1a). The dynamic gain was calculated by comparing eye-head velocities (see Methods). The gain was found to be 0.77 ± 0.19 for roll, 0.6 ± 0.23 for pitch, and 0.69 ± 0.13 for yaw. Position gain was retrieved by comparing the area-under-the-curve (AUC) between the eye and the head during the active movement. The gain values were 0.67 ± 0.16 for roll, 0.45 ± 0.16 for pitch, and 0.65 ± 0.25 for yaw.”

In the discussion we included the following sentence (line 379):

“The dynamic gain was recorded to be around 0.7 while the position gain approximated 0.6. These values are largely in line with previous studies on teleosts (Schairer and Bennett, 1986), while further trials are needed in order to further explore the lamprey VOR gain.”

We understand the concerns of the reviewer with respect to EMGs in extraocular muscles being a good indicator of eye movements. The reasons for using an isolated eye-brain-labyrinth preparation are exposed below in detail, and for using this approach we agree that is critical ensuring that normal eye movements are preserved. This is why we monitored eye movements with a video camera in response to vestibular stimulation, showing that VOR is behaviorally preserved in the preparation (Figure 1e, now in supplementary Fig. 1b). Although this shows that the system is functional, video tracking is not valid for quantifying vestibular-evoked eye movements in this preparation because the eye is outside its socket.

We therefore decided to use EMGs to reliably detect the changes in eye movement strength when applying the different paradigms. Our experience using similar preparations is that reliable eye movements are evoked both spontaneously and in response to electric stimulation. To show

this, we have performed new experiments electrically stimulating the AON vestibular nucleus at increasing intensities while recording extraocular muscle activity combined with video recordings of the evoked eye movements. These new results were incorporated in a new Supplementary Figure 1, and a new Supplementary Video (Supplementary Video 4) was added to aid visualization of how the activity recorded in the dorsal rectus corresponds to reliable eye movements. Supplementary Figure 1d shows that an increasing muscle activity corresponds to a larger amplitude of eye movements. In Supplementary Figure 1c and d it can also be seen that even a small activation of the muscle results in a detectable eye movement. These results show that, given the impossibility of doing these experiments in intact animals, the EMG activity that we show throughout the paper corresponds to eye movements of different amplitudes, and that the analysis of changes in this activity would correspond to changes in the amplitude of eye movements.

The following paragraph was included in the Results (line 125):

“To ensure that the EMG activity recorded in the extraocular muscles reliably corresponded to the eye movements, we performed electric stimulations at increasing intensities in the anterior octavomotor nucleus (AON; one of the vestibular nuclei involved in VOR, see below) while recording EMG activity in the dorsal rectus and simultaneously video recording eye movements (Supplementary Fig. 1c; N=2). The EMG activity increased in parallel with the amplitude of the eye movements (Supplementary Fig. 1c-e; Supplementary Video 4) and can thus be used to indirectly measure eye movements.”

In the Methods we included these experiments as follows (line 983):

“To ensure that normal eye movements were preserved in the preparation and that the recorded EMGs in the extraocular muscles corresponded to actual eye movements, we performed electric stimulation in the AON at increasing intensities, monitoring eye movements and simultaneously recording in the dorsal rectus (Supplementary Fig. 1).”

Figures 2, 3, 4. There are problems with the presentation of EMG responses. First, the EMG time courses are bursty and variable in amplitude. Normalisation of responses is therefore dangerous because it combines strong with weak responses without distinguishing between them. It would be better to show a mean unnormalised response along with either individual samples or confidence intervals. Second, the method of normalisation is unclear. Where are the voltage calibrations for individual responses? Why do points never reach a normalised value of 1 in most graphs? How is “amplitude” measured? The number of samples is also often

unclear. For example, where multiple recordings were made in the same animal, what was the total time interval, and were both sides of the animal used?

- Normalization: We agree with the reviewer that the explanation of the normalization method was not clear enough. We have now clarified this part in the methods (see below) and have also added a sentence in the text. As the reviewer states, there are strong and weak responses, but these differences are observed when comparing recordings from different animals. Thus, the aim of the normalization was indeed to ensure that such different responses are comparable among them.

In each animal we applied each paradigm (VIS, VES and VISVES) three to four times. As mentioned above, a difference in response strength was observed among animals, due to the excitability of each preparation that was variable, but also to the location of the electrode that gave rise to differences in signal strength among animals. However, the conditions were stable within each preparation and therefore the repetitions within each animal were very reproducible.

This means that responses recorded in one animal are well-suited for within-subject analyses, but the existing differences make normalization appropriate to compare responses between animals. Thus, what we did was to normalize the responses in each animal (dividing all values to the maximum response), so that for each animal we got values from zero to one. This strategy allowed us to compare data among animals, and what the graphs represent is the average percentage of the maximum response for each modality (VIS, VES, and VISVES). Given that what we look at is the ratio between sensory modalities and their relative influence in any given scenario, normalization enhances the visualization of this relationship.

We added the following sentence in the Results (line 168):

“As differences in signal strength were observed between animals, the responses were normalized in each animal to the maximum intensities allowing us to then compare the results obtained in the different preparations and to obtain the ratio between sensory modalities”

- Normalized values: Had the maximum response always been the one seen for VISVES, this value should be 1 in the graphs, as the reviewer notes. However, there were some cases where the maximum value was not VISVES but VES. Therefore, when averaging the data of all animals together (that were previously normalized as explained above) we got values lower than one.

- Voltage calibrations: We are sorry that that they were missing, and we have now included them.

- Amplitude measurements: Prior to quantifying amplitudes we first rectified the signals (making all negative values positive). The amplitude was then measured from the baseline to the maximum peak. We have now expanded the information in the methods as follows (line 1167):

“The signals were fully rectified and the distance from baseline to peaks was calculated as a way to measure the amplitudes.”

- Number of samples: To summarize the experimental design and respond to the questions addressed by the reviewer. Three/four recordings were made per animal, always on the same side (the electrode was placed at the beginning of the experiment at not moved until the end). After placing the electrode, the preparation was left to recover/adapt for at least one hour leaving the screens in the same white used as background for the stimuli. VIS, VES and VISVES trials were randomly presented to avoid adaptations and compensate for possible changes in the excitability of the preparation, and two minutes were left between trials. The total duration of the experiments not including the laborious dissection (from placement in the chamber to termination, including the recovering/adaptation time) was two and a half/three hours.

This information has been now expanded in the methods (line 1011).

“These paradigms were tested under four different conditions: low amplitude – low speed (5.8°; 48.7°/s), low amplitude – high speed (5.8°; 112.94°/s), high amplitude – low speed (22.7°; 48.7°/s), and high amplitude – high speed (22.7°; 112.94°/s). The angle refers to the maximum tilting of the platform, and the angular speed to the velocity of such tilting and/or the velocity of the optokinetic stimulation. Each of the paradigms (VIS, VES and VISVES) was applied three times on each animal under each condition (low amplitude – low speed, low amplitude – high speed, high amplitude – low speed, and high amplitude – high speed). The mean accelerations of the VES and VISVES protocols were 305.75 ± 112.79 deg/s² for the low velocity and 915.73 ± 324.58 deg/s² for the high velocity. This was calculated by attaching the accelerometer used in the behavioral studies to the platform and retrieving its motion dynamics. The recording electrode was placed in the dorsal rectus for roll and pitch recordings, and in the caudal rectus for yaw plane, and kept until termination of the experiment. After placing the electrode, the preparation was left to adapt for at least 30 min with a white screen

which was used as a background for all the applied stimuli. Two minutes were left between trials and presentation of VIS, VES and VISVES was randomized to minimize adaptation and compensate for possible changes in the excitability of the preparation. The number of repetitions performed on each animal was chosen to minimize the duration of the experiment while acquiring sufficient data, to ensure the viability of the preparation based on our experience in previous studies using similar isolated preparations. EMG responses to the specific protocols generally exhibited similar dynamics for each animal. This allowed us to identify trials in which the neural integrity of the preparation was in any way damaged, in which case the experiment would be terminated.”

Statistics. Statistical tests in the paper are frequently reported as a p-value with little further description. This is insufficient to judge the quality of the test and result. The following should be provided for every statistical test: name of test, test statistic, number of samples or degrees of freedom, the p-value.

We agree with the reviewer and have now included this information in the manuscript.

Role of pretectum in OKR. The authors maintain (line 23 of the Abstract) that the OKR is mediated by the pretectum. This assertion appears to rely on a single recording in a single animal – Figure 4h. This demonstration is inadequate. To support their claim, they need to show that the OKR is present before and after, but not during, pretectal inactivation in a statistically significant number of recordings across several animals.

We apologize for the oversight of not having included the number of animals (N=6) for this particular intervention and have now corrected this. The role of pretectum mediating OKR was tested by using both mechanical lesions (Figure 4g, 3 animals), and pharmacological inactivation (Figure 4h-i, 3 animals). Our initial decision to only show representative traces for each experiment stemmed from that the responses were essentially abolished in all cases after lesioning or injecting kynurenic acid in pretectum. We do however agree with the reviewer that this key result warrants some expansion, and we have performed a statistical analysis comparing responses before, during, and after injection of the drug to support the key role of pretectum mediating OKR. The new graph has been incorporated in the figure (Figure 4i), and appropriately referenced in the Results as follows (line 255):

“The plot in Fig. 4i shows that there is a significant reduction of OKR after KA injection in pretectum ($t(6) = 2.633$, $p = 0.036$), and that the OKR significantly recovers after washout ($t(3) = -3.703$, $p = 0.034$).”

Animal preparation. The study largely depends on an isolated eye-brain-labyrinth preparation. It was never clear to me why this was preferable to using a minimally invasive approach to the whole animal. Preparation of isolated tissues runs the risk of indirect damage to those tissues, and deterioration over time. The paper needs to justify the use of an isolated preparation, and to show that critical properties such as eyes movements were normal. The authors also need to indicate how they measured tissue deterioration over time, and criteria used to terminate an experiment.

We agree with the reviewer that working with intact animals would be preferential to using isolated preparations. We used this approach when possible, for example to prove VOR behaviorally, Figure 1, and to test nystagmus, Figure 6. However, it is extremely difficult to use this strategy to analyze visuovestibular integration. The long experiments make it impossible to keep a lamprey immobilized long enough to allow monitoring of eye movements while applying all the necessary paradigms, which we have learned through experience. We indeed tried several strategies to keep the head of the animal immobilized (wrap the neck with cloth, include it in agar, clamp it with rigid structures, etc.), but failed to produce a reliable methodology.

The most successful strategy (being as the reviewer mentions minimally invasive) was to place the animals in narrow chambers where they instinctively fixed their mouth to the chamber wall, so that we could take advantage of this static position to apply quick paradigms and test VOR and nystagmus behaviorally. Even for these short experiments, when the vestibular stimulation started in many cases the animals released the mouth moving the head, so the data shown in this work is the result of many repetitions until accumulating enough cases in which the animals did not move. However, as mentioned above, this strategy is not adequate to do longer experiments as the VISVES analysis.

Thus, we decided to use an isolated preparation, which also allowed us to perform recordings and inactivation of different brain areas to test their impact on eye movements as reflected in extraocular muscle activity, which would be impossible in intact animals. As the reviewer mentions, using an eye-brain-labyrinth preparation entails the risk of perturbing the natural eye movements of the animal. However, as explained in the above comment, the registered extraocular muscle activity is the result of reliable eye movements in this preparation and, given the impossibility of getting actual eye movements in intact animals, EMG activity is therefore a good strategy to allow the analysis of visual and vestibular contributions. Naturally,

a study investigating the muscular dynamics of the eye movement would benefit from trials in animals with intact structures, and as this study focuses on the central sensory integration, we have investigated the neural pathways and used the muscle activity as an indicator of the desired motor output, although not necessarily the optimal one.

We agree that we need to justify using this preparation, and we have therefore included this sentence in the second paragraph of the Results sections (line 104):

“In intact animals we could investigate the VOR behaviorally, but analyzing the contribution of visual and vestibular information, which entailed long experiments, was not possible to do in intact animals (see Methods).”

In the Methods, we have included the following sentences to justify the usage of this preparation (line 966):

“Placing animals in a transparent chamber generally saw them instinctively attaching their mouths to the chamber wall, stabilizing their head and allowing us to test VOR and nystagmus behaviorally. However, analyzing the relative contribution of visual and vestibular information required applying several experimental paradigms and could not be performed on intact animals. Thus, we decided to use an isolated preparation of the lamprey brain and rostral spinal cord with the eyes and vestibular organs. This allowed us to monitor eye movements via EMG recordings of the extraocular muscles to test visuovestibular integration, as well as record and inactivate different brain regions.”

Regarding tissue deterioration, we have used lamprey *ex vivo* preparations maintaining the eyes and brain intact in several studies (Pérez-Fernández et al., 2017; Suzuki et al., 2019; Suryanarayana et al., 2020), and our experience is that reliable neuronal activity can be recorded for many hours (we sometimes did experiments during seven-eight hours), and eye movements are observed, both spontaneous and evoked by electric stimulation. Moreover, eye movements could be observed the next day in preparations that were injected with neuronal tracers and left overnight in aCSF to allow transport. Although the duration of the experiments carried out in this study was much shorter than this, we considered tissue deterioration, which was evaluated through visually monitoring any changes in the eye movement responses over. As outlined above, the EMG activity for each animal was very consistent, so in the few cases in which a remarkable reduction in EMG activity was observed, the experiment was terminated. In this sense, the number of repetitions performed in each animal was chosen to fall in a safe duration range according to our previous experience so that

the preparation was in good conditions. To explain this in the Methods, we have inserted the following sentences (line 1025):

“The number of repetitions performed on each animal was chosen to minimize the duration of the experiment while acquiring sufficient data, to ensure the viability of the preparation based on our experience in previous studies using similar isolated preparations (Pérez-Fernández et al., 2017; Suzuki et al., 2019; Suryanarayana et al., 2020).”

Methods. The Methods section is often unclear. For example, lines 815 and 854 state that “all muscles were removed”. How, then, were EMGs recorded? I suggest that at least two of the authors review this section to improve its clarity.

We agree that this section was in need of clarification on several points and have now incorporated numerous additions to improve the readability of the Methods section. To simplify this response letter, we will not include these different modifications here, but all the changes done are indicated in red in the track-changes revised manuscript. Regarding muscles, this sentence has now been rephrased to:

“...all muscles except the extraocular were removed”.

There were also some less serious problems with the paper, as follows.

- Line 968. Outliers were removed. This is dangerous because examination of the outliers could reveal previously unappreciated factors in the experiment. Outliers should be described, and their removal should be justified.

Using a preparation that maintains the whole brain together with the eyes means that spontaneous eye movements were present. There were some cases where eye movements were produced that fell in the time frame of the analysis, or immediately before the stimulation, so that they generated data that may at first have appeared to be a response, but upon closer inspection during the in-depth analysis proved to be different from the other responses, showing a rapid spasm incompatible with a functional gaze-stabilization. We first checked these outliers visually and, after realizing their nature, decided to apply a Grubb's test to identify significant outliers, and those removed were within a 95% confidence interval. While one may argue that these outliers could have been excluded from the data synthesis, we opted to test their validity in a more objective way, hence their removal during the Grubb's test. To describe the nature of these outliers, we have now expanded this sentence in the Methods (line 1185):

“In some cases, spontaneous eye movements were generated immediately before or after the stimulation, clearly affecting the responses analyzed. After visually inspecting putative outliers, a Grubb’s test was used to identify significant outliers, and data points outside of the 95% confidence interval were removed prior to performing the statistical analysis”

- Vestibular stimulation is transient. Given that a change from one rotation speed to another cannot be instantaneous, the time course of acceleration should be described.

We agree and have now included the acceleration values that are indicated in the methods as follows (line 1016):

“The mean accelerations of the VES and VISVES protocols were $305.75 \pm 112.79 \text{ deg/s}^2$ for the low velocity and $915.73 \pm 324.58 \text{ deg/s}^2$ for the high velocity. This was calculated by attaching the accelerometer used in the behavioral studies to the platform and retrieving its motion dynamics.”

The type of servo motor used functions by accelerating to maximum to reach as soon as possible the desired velocity.

- A positive note! The nystagmus in Figure 6 is fascinating, even more so because it was shown in the whole animal. How does the speed of the quick phase compare with that in other cold-water animals, and how well does the eye movement compensate for head rotation?

Thank you, as mentioned above we managed to implement an experimental setup that allowed us to perform quick experiments so that animals could be kept static with a minimally invasive approach for testing this. Regarding gain, we have now performed new experiments as exposed above, showing that the dynamic gain is around 0.7, and the static gain around 0.6. For quick phases, as suggested by reviewer three, we have now included their amplitudes and velocities.

We have now included the following sentence in the text discussing the lamprey quick phase velocity with that of fish (line 438):

“The velocity of saccadic movements in fish ranges between a hundred and a few hundreds of degrees per second (Easter et al., 1997; Fernald, 1985; Easter, 1971). Thus, saccadic eye movements are slightly slower in lampreys. In the lamprey, a functional cerebellum has not evolved, and the VOR can thus not have the complementary cerebellar circuit, which in other vertebrates is used for fine tuning of the VOR.”

Reviewer #3 (Remarks to the Author):

This manuscript examines the visual mechanisms of gaze-stabilization in the lamprey, a member of the most ancient group of vertebrates. Using an ingenious platform to conduct electrophysiological recordings while measuring eye movements and providing vestibular and visual stimulation, this study investigates the influence of visual signals on the vestibulo-ocular reflex and the underlying neural pathways. Behavioral data show visuomotor responses and a response pattern reminiscent of optokinetic nystagmus. Electrophysiological results show strong visuo-vestibular integration and identify various areas involved in this process. I find these experiments well designed and the results interesting. There are, however, various points in the text in which things are not well explained and need to be refined.

We thank the reviewer for the kind comments and the useful suggestions.

General comments

Nystagmus. One aspect that is not provided in the manuscript is a characterization of the type of behavior elicited by visual stimulation. Part of the problem in understanding this is also that the text does not clearly explain in which way the stimulus and the platform move. What is causing the cyclical behavior of the traces in Figure 1, a sinusoidal oscillation of the platform or a sort of optokinetic nystagmus? One would assume a OKN-type of behavior to be visible in the absence of VOR, and perhaps this is the function of Figure 1a. But things remain unclear both because the experimental conditions are not well explained and because examination of nystagmus is postponed to much later (Figure 6), using a different preparation. In addition to clarifying these issues, it would be helpful to see if the resetting movements---with or without VOR---do resemble saccades, by reporting their main sequence characteristics (the relationships between peak speed, duration, and amplitude).

We agree that the figure traces resemble those that may be seen during an OKR, but in this case we are showing nystagmus evoked only by vestibular stimulation (VOR nystagmus). In Figure 1, as the reviewer says, responses are the result of a vestibular sinusoidal stimulation, with the experiments performed in darkness to avoid optokinetic responses. We also agree that the Methods were not clear enough in some cases, so we have now improved them with several additions (see previous responses). Some of the changes have already been included in relation to specific comments, but others we are not including here to facilitate the lecture of this response letter. However, all of them are indicated in the track changes manuscript.

Nystagmus analysis had to be done in intact animals, and, as described in the responses to Reviewer 2, it is difficult to keep the lampreys' head immobilized, so we took advantage of

their instinct to attach their mouth by using narrow transparent chambers. We opted not to expand upon the nature of any potential nystagmus movements in the dissected preparation, as we could determine their main features from intact animals.

We agree that a more detailed description of the resetting movements is important, and we have now included the velocity and amplitude, apart from the previously reported duration, in the text as follows (line 326):

“...while the duration of the resetting phase yielded very similar durations (0.024 ± 0.009 seconds) across different stimulation velocities in a fashion typical of a quick-phase nystagmic eye movement (Fig. 6d), showing that movements of saccadic nature are present in the lamprey. The analysis of these resetting eye-movements dynamics revealed that their velocity was 77.06 ± 30.30 deg/s and the amplitude $1.72 \pm 0.73^\circ$, with no significant differences when comparing quick phases evoked at different stimulation velocities”.

As indicated in the previous comment to Reviewer 2, we have also included a sentence in the Discussion.

“The velocity of saccadic movements in fish ranges between a hundred and a few hundreds of degrees per second (Easter et al., 1997; Fernald, 1985; Easter, 1971). Thus, saccadic eye movements are slightly slower in lampreys. In the lamprey, a functional cerebellum has not evolved, and the VOR can thus not have the complementary cerebellar circuit, which in other vertebrates is used for fine tuning of the VOR.”

Corollary discharges and yaw movements. The section on lines 292-314 is very confusing. To start, the driving hypothesis is not clear: why the presence of non-visually driven compensatory eye movements would justify the absence of OKR on the yaw plane? The conclusions seem contradictory: it is first stated that compensatory movements were only observed in one out of four animals (line 302; why proceeding if this behavior is not typical?). And later concluded that no compensatory responses were evoked in the eye muscles (line 312; not even in the one animal that did exhibit a compensatory behavior? How could this be?). I must be missing something here, because the entire section does not seem to make much sense.

As commented by the reviewer, behaviorally speaking our results suggest that corollary discharges alone cannot generate eye movements in most cases. However, in one animal we clearly obtained eye movements coordinated with the tail that persisted when the visual and vestibular systems were inactivated (Figure 6). The only possible explanation for these results

is the presence of corollary discharges evoked by locomotion. Given the difficulties to get these eye movements in semi-intact preparations, we decided to use an ex-vivo preparation evoking locomotion with glutamate. Apart from having a more controlled conditions (we could very reliably evoke long swimming episodes), the logic of these experiments was that perhaps some weak muscle activity not reflected in visible eye movements could be recorded. However, this was not the case as shown in supplementary Figure 5.

Altogether, the presence of locomotion-evoked corollary discharges is clear, although their contribution seems to be largely subthreshold. To analyze the contribution of corollary discharges and their underlying synaptic mechanisms, our long-term goal is to perform intracellular recordings in oculomotor neurons in response to locomotion, but this project is technically very challenging and will therefore become a full separate study. However, given that we show the main components that participate in gaze stabilization in lampreys, we think it is appropriate to include these results.

To clarify this section, we have now added several sentences in the text:

Second paragraph in this section of the Results (line 348):

“Given that clear compensatory eye movements coordinated with locomotion were observed, albeit only in one animal, we investigated whether small activation of the extraocular eye muscles that did not result in perceivable eye movements could be recorded via EMGs.”

The following sentence: “Altogether, these results show that corollary discharges can generate compensatory eye movements, although its contribution is in most cases subthreshold.” Was rephrased to (line 358): “Altogether, these results show that corollary discharges can generate compensatory eye movements, as clearly observed in one animal, although its contribution is in most cases subthreshold or absent, and the origin remains to be determined.”

We also realized that the title of supplementary figure 5 was wrong, increasing to the confusion. We have now changed it from “Compensatory eye movements with intact labyrinths” to (line 904) “Fictive locomotion does not result in coordinated extraocular muscle activity”

Figures. Another general comment regards the description of the figures, which are overall nice, but often contain elements that are not explained neither in the text nor caption. Several specific examples are provided below.

Specific comments

Figure 1. (1A) What are the two columns of eye position? Please insert labels on top. And the circles and dashed lines in each panel? (1F) I find the way of marking visual stimulation via stripes difficult to read, as it overlaps with the data and other event markers. Why not adding temporal sequence of events as bars on top/bottom?

The two images of the eye show the initial position (before vestibular stimulation), and at the peak of vestibular stimulation. To clarify this, we have now inserted labels as suggested by the reviewer.

The circles are the labels used by the eye tracking system (DeepLabCut). We have expanded the information in the figure legend from “Labels used to track eye trajectories can be seen as color dots” to (line 643) “the colored dots in the images represent the labels used for the tracking system (DeepLabCut) to calculate the eye trajectory.”

The lines indicate the initial position of the eye to be compared with the position after the vestibular stimulation. To clarify this, we have now added lines in green to also have the final position, and the information was added in the figure legend (line 642):

“The red dotted line indicates the position of the center of the eye before stimulation, and the green dotted line its position at the peak of vestibular stimulation. (bottom).”

Regarding the way of marking visual stimulation, we agree that the oblique bars on top make it difficult to interpret the figures. We tried with bars as suggested by the reviewer, but in some cases it was not obvious if they were referring to the traces on top or below, given the structure of the figures. Thus, we chose to use a dashed rectangle that, in our opinion, is very intuitive and does not overlap with other events marks. All figure legends were corrected to indicate this change.

Figure 2. Why are EMG amplitudes and spikes plotted in normalized coordinates? The text does not explain what information is gained from either of these two metrics individually. The amplitude is relegated to a supplementary figure in Figure 3, but not clear why. Simply because it didn't yield a significant result?

We here expose the reasons for normalization also explained in another comment above: In each animal we applied each paradigm (VIS, VES and VISVES) three to four times. A difference in response strength was observed among animals, due to differences in the excitability of each preparation, but also to the location of the electrode that gave rise to differences in

signal strength among animals. However, the conditions were stable for each preparation and therefore the repetitions within each animal were very reproducible.

Thus, the responses recorded in one animal are well-suited for within-subject analyses, but the existing differences make normalization appropriate to compare responses between animals. Thus, we normalized the responses in each animal (dividing all values to the maximum response), so that for each animal we got values from zero to one. This strategy allowed us to compare data among animals and, given that what we look at is the ratio between sensory modalities and their relative influence in any given scenario, normalization enhances the visualization of this relationship.

The information from each metric would be the amplitude of the eye movements (reflected in EMG amplitude), and the speed/duration of the eye movement (reflected in the frequency/duration of spikes). As mentioned above, it was not our aim to extract the kinematics of eye movements which is hard with this type of data. However, we think that this analysis demonstrates that visual information impacts on both the number of evoked spikes and maximum EMG amplitude, suggesting that vision influences the amplitude, speed, and duration of eye movements.

Regarding the EMG amplitudes data, it was moved to supplementary just for a matter of space, same as the traces in response to low amplitude high speed and high amplitude low speed stimulations. Visuovestibular stimulation yielded significant differences compared to only vestibular for low amplitude high speed stimulation, and for high amplitude low speed stimulation. We have now moved the amplitude graphs to the main figure and kept the low amplitude high speed stimulation and high amplitude low speed stimulation traces in the supplementary figure.

Figure 3. The characteristics of motion are not clear. I assume that here amplitude and speed refer to the peak angle of the platform and speed of motion of the visual stimulus, but the text does not explain. Part of the problem is that I would expect the platform to continually rotate to study VOR, but panel F seems to state that the platform only moves briefly at the very beginning (the blue stripe; note that legends are not defined here). Is the platform moving or not?

As the reviewer says, the amplitude refers to the peak angle of the platform, and the speed to the speed of motion of the platform and/or the visual stimulus. We have now included the following sentence in the figure legend to clarify this (line 693):

“Amplitude refers to the peak tilting angle of the platform, and speed to the motion velocity of the platform or the visual stimulus”

The vestibular stimulation is transient, and both in the figure and in the figure legend we indicate that blue corresponds to the duration of the roll movement, and yellow indicates that the platform stays at a tilted position. Our goal was to see whether visual and vestibular inputs interact, which the short duration of stimulation allowed us to do. As commented by the reviewer, it would be also interesting to apply a more prolonged stimulation to extract more details about how visual and vestibular inputs interact through time. Although this is hard to achieve in the roll plane with this experimental preparation, we are working on this type of experiments for future studies.

A further thing to pay attention to is the use of the term amplitude to describe the movement characteristics, creating possible confusion with the amplitude of the EMG signals.

We have now changed “amplitude” to “eye movement amplitude” or “EMG amplitude” where relevant.

Line 162. Not clear why the authors conclude that visual stimulation plays a more important role at the beginning of compensatory eye movements.

This conclusion was based on the notion that visual stimulation has a greater impact during low amplitude stimulations (this is known for humans; Wibble et al., 2020), and we show here that the VISVES number of spikes is significantly different from VES for both low and high speeds, and the VISVES EMG amplitude is significantly different from only VES for low amplitude high speed stimulation). However, for high amplitude stimulation the visual contribution is only significant for EMG amplitudes at low speed. On the other hand, as reflected both in the raw traces of Figure 3j-k and supplementary Figure 2, and in the heatmaps of Figure 3p, suprathreshold responses are generally evoked at the beginning of visual stimulation. While optokinetic stimulation also has an impact prolonging the responses (as seen in the heatmaps), these two aspects suggest that its role is more prominent at the beginning of the movements. We have expanded this sentence at the end of the section to (line 215):

“The above results show that joint visuo-vestibular stimulations enhance the responses, but the visual impact is larger at low amplitudes movements. This finding, together with that the suprathreshold VIS responses are evoked only at the beginning of the visual stimulation, suggests that its role is particularly important at the beginning of compensatory eye movements.”

Cited reference: Wibble, T., Engström, J. & Pansell, T. Visual and Vestibular Integration Express Summative Eye Movement Responses and Reveal Higher Visual Acceleration Sensitivity than Previously Described. *Invest. Ophthalmol. Vis. Sci.* **61**, 4 (2020).

Line 223: Supplementary Figure 3i not 3b

The reviewer is correct, and we have now changed this.

Line 268: Maybe Figure 5i, rather than 5g?

As the reviewer has correctly pointed out, 5g is wrong. We here depolarized the neurons to -50 mV so that action potentials could be observed, and no cessation or reduction of firing was observed after stimulating pretectum, suggesting that there is no inhibition as proved after intracellular blockade of sodium channels (Supplementary Fig. 4b). We have consequently corrected the reference to 5h.

REVIEWERS' COMMENTS

Reviewer #1 (Remarks to the Author):

This paper by Perez-Fernandez is presenting for the first time the role of vestibular-ocular interactions. The data are well presented and have now all relevant papers are cited. Overall, the VOR shows a similar function in lampreys despite the absence of a horizontal canal. The methodology is sound and the discussion is clear.

Reviewer #2 (Remarks to the Author):

The authors have addressed my concerns.

Reviewer #3 (Remarks to the Author):

The authors have satisfactorily responded to my comments. I think the article has improved and provides a nice contribution to the literature.

While I do not have additional substantive comments, I would recommend further revising the Introduction, which remains a bit meandering. This section could be shortened and made it to address more directly the relevant questions. Make sure also to fix typos and repetitions (e.g., “although and although” on line 50, “well-developed” on line 59, etc.)

I still do not understand the point of the section on locomotion, the last component of the Results. Given that these results are inconsistent across samples, it is difficult to draw any conclusions. It seems that this is something that can be mentioned in the Discussion and moved to supplementary information, given that it adds very little to the article.

Reviewer #1 (Remarks to the Author):

This paper by Perez-Fernandez is presenting for the first time the role of vestibular-ocular interactions. The data are well presented and have now all relevant papers are cited. Overall, the VOR shows a similar function in lampreys despite the absence of a horizontal canal. The methodology is sound and the discussion is clear.

We thank the reviewer for the kind words and helpful suggestions.

Reviewer #2 (Remarks to the Author):

The authors have addressed my concerns.

Thank you for the suggested experiments that, we think, have improved the quality of this work.

Reviewer #3 (Remarks to the Author):

The authors have satisfactorily responded to my comments. I think the article has improved and provides a nice contribution to the literature.

While I do not have additional substantive comments, I would recommend further revising the Introduction, which remains a bit meandering. This section could be shortened and made it to address more directly the relevant questions. Make sure also to fix typos and repetitions (e.g., “although and although” on line 50, “well-developed” on line 59, etc.)

I still do not understand the point of the section on locomotion, the last component of the Results. Given that these results are inconsistent across samples, it is difficult to draw any conclusions. It seems that this is something that can be mentioned in the Discussion and moved to supplementary information, given that it adds very little to the article.

We are happy that our amendments have been well-received and are thankful for the opportunity to improve on the quality of the article during the revision. We will now prepare the manuscript for publication and go through the text to remedy any remaining linguistic oversights that have remained. To this effect we are grateful for the examples highlighted in the reviewer’s comment.

Regarding the inclusion of locomotion in the main text, we agree with the reviewer in that it is a matter worthy of some consideration. After much discussion, we included this segment in the main text after having first drafted it as part of the supplementary, as suggested in this comment. We however felt that it was worthy of inclusion in the end, as it constitutes an important role in eye-movement control that is well-documented but quite often overlooked. By including it in the main text we feel quite strongly that the manuscript offers a more representative description of vertebrate gaze control, and that it creates a broader platform from which future studies may be launched. We are thankful for having been given the opportunity to expand on our reasoning for this inclusion in the manuscript, and hope that the reviewer may sympathize with our perspective.